# TOWARDS MITIGATING ARCHITECTURE OVERFITTING IN DATASET DISTILLATION

## ABSTRACT

Dataset distillation methods have demonstrated remarkable performance for neural networks trained with very limited training data. However, a significant challenge arises in the form of *architecture overfitting*: the distilled training data synthesized by a specific network architecture (i.e., training network) generates poor performance when trained by other network architectures (i.e., test networks). This paper addresses this issue and proposes a series of approaches in both architecture designs and training schemes which can be adopted together to boost the generalization performance across different network architectures on the distilled training data. We conduct extensive experiments to demonstrate the effectiveness and generality of our methods. Particularly, across various scenarios involving different sizes of distilled data, our approaches achieve comparable or superior performance to existing methods when training on the distilled data using networks with larger capacities.

## 1 INTRODUCTION

Deep learning has achieved tremendous success in various applications (Rombach et al., 2022; Jumper et al., 2021), but training a powerful deep neural network requires massive training data (Dosovitskiy et al., 2020; Brown et al., 2020). To accelerate training, one possible way is to construct a new but smaller training set that preserves most of the information of the original large set. In this regard, we can use *coreset* (Coleman et al., 2019; Hwang et al., 2020) to sample a subset of the original training set or *dataset distillation* (Wang et al., 2018; Zhao et al., 2020) to synthesize a small training set. Compared to coreset, dataset distillation achieves much better performance when the amount of data is extremely small (Hwang et al., 2020; Zhao & Bilen, 2023). Furthermore, dataset distillation is shown to benefit various applications, such as continual learning (Zhao et al., 2020; Rosasco et al., 2022; Zhao & Bilen, 2021; 2023), neural architecture search (Zhao et al., 2020; Zhao & Bilen, 2021), and privacy preservation (Li et al., 2020; Goetz & Tewari, 2020). Therefore, in this work, we focus on dataset distillation to compress the training set.

In the dataset distillation framework, the small training set, which is also called the *distilled dataset*, is learned by using a neural network (i.e., training network) to extract the most important information from the original training set. Existing data distillation methods are based on various techniques, including meta-learning (Wang et al., 2018; Bohdal et al., 2020; Sucholutsky & Schonlau, 2021; Nguyen et al., 2021; Zhou et al., 2022) and data matching (Zhao et al., 2020; Cazenavette et al., 2022; Cui et al., 2022; Wang et al., 2022; Zhao & Bilen, 2023). These methods are then evaluated by the test accuracy of another neural network (i.e., test network) trained on the distilled dataset. Despite efficiency, dataset distillation methods generally suffer from *architecture overfitting* (Zhao & Bilen, 2021; 2023; Nguyen et al., 2021; Zhou et al., 2022; Cazenavette et al., 2022). That is, the performance of the test network trained on the distilled dataset degrades significantly when it has a different network architecture from the training network. Moreover, the performance deteriorates further when there is a larger difference between the training and test networks in terms of depth and topological structure. Due to high computational complexity and optimization challenges in dataset distillation, the training networks are usually shallow networks, such as 3-layer convolutional neural networks (CNN) (Cazenavette et al., 2022; Zhou et al., 2022). However, such shallow networks lack representation power in practical applications. In addition, deep networks have shown stronger representation power in many tasks (He et al., 2016; Dosovitskiy et al., 2020). Therefore, we believe a deeper network has the potential for better performance when trained on distilled datasets.

Note that, our analysis indicates that the performance gap between different network architectures is larger in the case of training on the distilled dataset than in the case of training on the subset of the original training set. In addition,compared with methods compressing the training set by subset selection, dataset distillation achieves better performance when using the same amount of training instances and is thus more popular in downstream applications (Zhao & Bilen, 2021; 2023; Goetz & Tewari, 2020). Therefore, we focus on dataset distillation, in which the effectiveness of the proposed method can be better revealed.

In this work, we demonstrate that the architecture overfitting issue in dataset distillation can be mitigated by a better architecture design and training scheme of test networks on the distilled dataset. We propose a series of approaches to mitigate architecture overfitting in dataset distillation. Specifically, these approaches can be categorized into four types: **a) architecture:** DropPath with three-phase keep rate and improved shortcut connection; **b) objective function:** knowledge distillation from a smaller teacher network; **c) optimization:** periodical learning rates and a better optimizer; **d) data:** a stronger augmentation scheme. Our proposed methods are generic: we conduct comprehensive experiments on different network architectures, different numbers of instances per class (IPC), different dataset distillation methods and different datasets to demonstrate the effectiveness of our methods. Figure 1 below demonstrates the performance of our proposed methods in various scenarios. It is clear that our methods can greatly mitigate architecture overfitting and make large networks achieve better performance in most cases. In addition to dataset distillation, our methods can also improve the performance of training on a small real dataset, including those constructed by corsets. Although some tasks, like synthetic-to-real generalization (Chen et al., 2021) and few-shot learning (Parnami & Lee, 2022), are also classical problems, customizing our method for them is out of the scope of this work, because we focus on training large networks on small datasets from scratch. We leave these tasks as future works. What's more, compared with the existing methods, our proposed methods introduce negligible overhead and are thus computationally efficient.

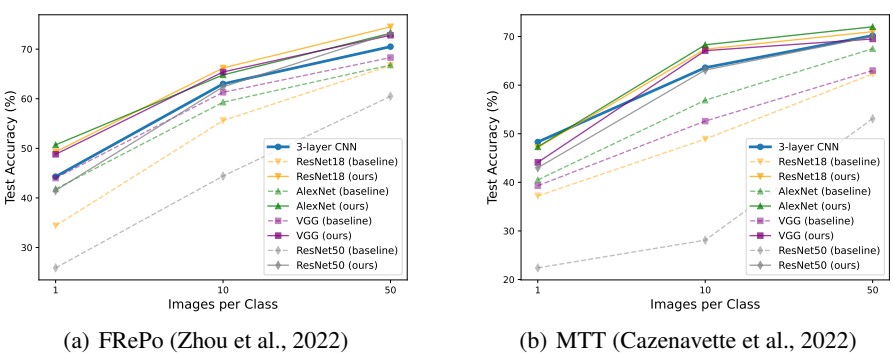

(a) FRePo (Zhou et al., 2022)  (b) MTT (Cazenavette et al., 2022)

Figure 1: Effectiveness of our method on different architectures, different dataset distillation methods, and different images per class (IPCs) on CIFAR10. We use a 3-layer CNN as the training network, so it performs the best among various architectures under baselines (dashed lines). Our methods (solid lines) can significantly narrow down the performance gap between the 3-layer CNN and other architectures.

We summarize the contributions of this paper as follows:

1. We propose a series of approaches to mitigate architecture overfitting in dataset distillation. They are generic and applicable to different model architectures and training schemes.

2. We conduct extensive experiments to demonstrate that our method significantly mitigates architecture overfitting for different network architectures, different dataset distillation approaches, different IPCs, and different datasets.

3. Moreover, our method generally improves the performance of deep networks trained on limited real data. As a result, deep networks outperform shallow networks on different fractions of training data, even when there are only 100 training samples.

## 2 RELATED WORKS

**Dataset Distillation:** The goal of dataset distillation is to learn a smaller set of training samples (i.e. distilled dataset) that preserves essential information of the original large dataset so that the model trained on this small dataset performs similarly to that trained on the original large dataset. Existing dataset distillation approaches are based on either meta-learning or data matching (Lei & Tao, 2023). The former category includes backpropagation through time (BPTT) approach (Wang et al., 2018; Bohdal et al., 2020; Sucholutsky & Schonlau, 2021) and kernel ridge regression (KRR) approach (Nguyen et al., 2021; Zhou et al., 2022); the latter category includes gradient matching (Zhao & Bilen, 2021; Lee et al., 2022b), trajectory matching (Cazenavette et al., 2022; Du et al., 2022; Cui et al., 2022), and distribution matching (Wang et al., 2022; Zhao & Bilen, 2023). However, these methods suffer from severe architecture overfitting, which means significant performance degradation when the architecture of the training network and the test network are different. Recently, some factorization methods (Kim et al., 2022; Deng & Russakovsky, 2022; Liu et al., 2022; Lee et al., 2022a), which learn synthetic datasets by optimizing their factorized features and corresponding decoders, greatly improve the cross-architecture transferability. However, the instance per class (IPC), which indicates the number of instances in the distilled dataset, used in these methods is at least 5 times larger than that of meta-learning and data matching approaches, which greatly cancels out the advantages of dataset distillation. To better fit the motivation of dataset distillation, we only consider small IPCs (1, 10 and 50) in this work, so the factorization methods are not included for comparison.

**Model Ensemble:** Model ensemble aims to integrate multiple models to improve the generalization performance. Popular ensemble methods for classification models include bagging (Breiman, 1996), AdaBoost (Hastie et al., 2009), random forest (Breiman, 2001), random subspace (Ho, 1995), and gradient boosting (Friedman, 2002). However, these methods require training several models and thus are computationally expensive. By contrast, DropOut (Srivastava et al., 2014) trains the model only once but stochastically masks its intermediate feature maps during training. At each training iteration with DropOut, only part of the model parameters are updated, which forms a sub-network of the model. In this regard, DropOut enables implicit model ensembles of different sub-networks to improve the generalization performance. Similar to DropOut, DropPath (Larsson et al., 2016) also implicitly ensembles sub-networks but it blocks a whole layer rather than masking some feature maps. Therefore, it is applicable to network architectures with multiple branches, such as ResNet (He et al., 2016), otherwise, the model output will be zero if a layer of a single branch network is dropped. By contrast, we propose a DropPath variant in this work which is generic, applicable to single-branch networks and effective to mitigate architecture overfitting.

**Knowledge Distillation:** Knowledge distillation (Hinton et al., 2015) aims to compress a well-trained large model (i.e., teacher model) into a smaller and more efficient model (i.e., student model) with comparable performance. The standard knowledge distillation (Hinton et al., 2015) is also known as offline distillation since the teacher model is fixed when training the student model. Online distillation (Zhang et al., 2018; Chen et al., 2020a) is proposed to further improve the performance of the student model, especially when a large-capacity high-performance teacher model is not available. In online distillation, both the teacher model and the student model are updated simultaneously. In most cases, knowledge distillation methods use large models as the teachers and small models as the students, which is based on the fact that larger models typically have better performance. However, in the context of dataset distillation, a smaller test network with the same architecture as the training network can achieve a better performance than a larger one on the distilled dataset, so we use the small model as the teacher and the large model as the student in this work.

We show in the following sections that combining DropPath and knowledge distillation, architecture overfitting in dataset distillation can be almost overcome.

## 3 METHODS

In this section, we introduce the approaches that are effective in mitigating architecture overfitting in dataset distillation. Our methods are motivated by traditional wisdom to mitigate overfitting, including ensemble learning (Ueda & Nakano, 1996; Srivastava et al., 2014), regularization (Krogh & Hertz, 1991; Chen et al., 2020b) and data augmentation (Halevy et al., 2009; Shorten & Khoshgoftaar, 2019). First, we propose a DropPath variant, which implicitly ensemble subsets of models and

is different from vanilla DropPath (Larsson et al., 2016) in that it is also applicable to single-branch architectures. Correspondingly, we optimize the shortcut connections of ResNet-like architecture to better accommodate DropPath. Second, we use knowledge distillation (Hinton et al., 2015) as a form of regularization to improve the performance to a large extent, even though the teacher model is actually smaller than the student model in our cases. Finally, we adopt a periodical learning rate scheduler, a gradient symbol-based optimizer (Chen et al., 2023), and a stronger data augmentation scheme when training models on the distilled dataset, to further improve the performance.

## 3.1 DROPPATH

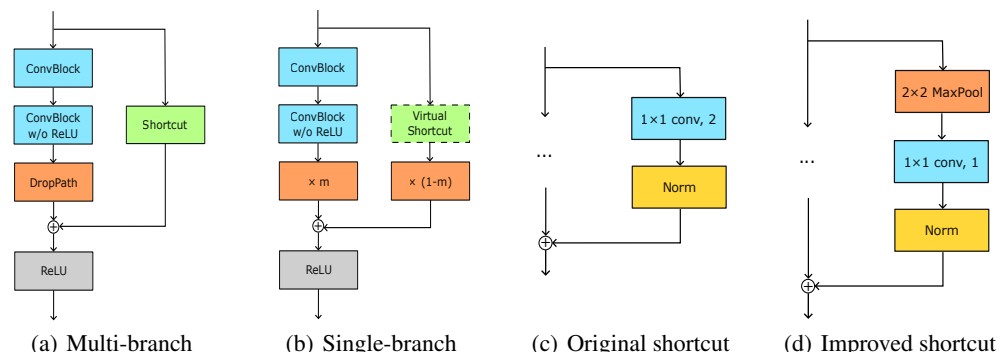

(a) Multi-branch     (b) Single-branch     (c) Original shortcut     (d) Improved shortcut

Figure 2: **(a)** The DropPath used for multi-branch residual blocks during training, it does not block the shortcut path. **(b)** The DropPath used for single-branch networks during training. Here, $m = \texttt{Bernoulli}(p) \in \{0, 1\}$, $p \in [0, 1]$ denotes the keep rate. Only when the main path is pruned ($m = 0$), the virtual shortcut is activated, and vice versa. DropPath is always deactivated, i.e., $p = 1$, during inference. **(c)** The original architecture of a shortcut connection to downsample feature maps, which consists of a $1 \times 1$ convolution layer with the stride of 2 and a normalization layer. **(d)** The improved architecture of a shortcut connection to downsample feature maps, which is a sequence of a $2 \times 2$ max pooling layer, a $1 \times 1$ convolution layer with the stride of 1 and a normalization layer.

Similar to DropOut (Srivastava et al., 2014), DropPath (Larsson et al., 2016), a.k.a., stochastic depth, was proposed to improve generalization. While DropOut masks some entries of feature maps, Drop-Path randomly prunes the entire branch in a multi-branch architecture. To obtain a deterministic model for evaluation, DropPath is deactivated during inference. To ensure the expectation of the feature maps to be consistent for training and inference, we scale the output of feature maps after DropPath during training. Mathematically, DropPath works as follows:

$$\texttt{DropPath}(\mathbf{x}) = \frac{m}{p} \cdot \mathbf{x}, \quad m = \texttt{Bernoulli}(p), \tag{1}$$

where $p \in [0, 1]$ denotes the keep rate, $m = \texttt{Bernoulli}(p) \in \{0, 1\}$ outputs 1 with probability $p$ and 0 with probability $1 - p$. The scaling factor $1/p$ is used to ensure the expectation of the feature maps remains unchanged after DropPath. Detailed derivation is in Appendix A.2. Figure 2 (a) illustrates how DropPath is integrated into networks. It effectively decreases the model complexity during training and can force the model to learn more generalizable representations using fewer layers. Same as DropOut, any network trained with DropPath can be regarded as an ensemble of its subnetworks (Krizhevsky et al., 2012). Ensembling has been proven to improve generalization (Breiman, 1996; Hastie et al., 2009; Breiman, 2001; Ho, 1995; Friedman, 2002). As a result, we can also expect DropPath to mitigate the architecture overfitting issue in dataset distillation. Note that, DropOut masks part of the feature maps and effectively decreases the network width; by contrast, DropPath removes a branch and thus decreases the network depth. Architecture overfitting arises from deeper test networks, so we use DropPath instead of DropOut in this context.

**Three-Phase Keep Rate:** The keep rate $p$ is the key parameter that controls the effective depth of architecture. Since the architecture factor $m = \texttt{Bernoulli}(p)$, the effective depth gets smaller as $p$ decreases. In the early phase of training, the model is underfitting, stochastic architecture brings optimization challenges for training, so we turn off DropPath by setting the keep rate $p = 1$ in the first few epochs to ensure that the network learns meaningful representations. We then gradually

decrease $p$ to decrease the effective depth and thus to narrow down the performance gap between the deep sub-network and the shallow training network until it reaches the predefined minimum value after several epochs. In the final phase of training, we decrease the architecture stochasticity by increasing the value of $p$ to a higher value to ensure training convergence. In experiments, we shrink the keep rate every few epochs. The pseudo-code is shown in Algorithm 1 of Appendix A.1. Figure 6 of Appendix A.1 illustrates the scheduler of the keep rate.

**Generalize to Single-Branch Networks:** Since DropPath prunes the entire branch, it is not applicable to single-branch networks, such as VGG (Simonyan & Zisserman, 2014). This is because we need to ensure the input and the output of the network are always connected, otherwise, we will obtain a trivial constant model. In the case of ResNet, we prune the main path of a residual block stochastically, while the shortcut connections are always reserved.

To improve the performance of single-branch networks, we propose a variant of DropPath. As illustrated in Figure 2(b), we add a virtual shortcut connection between two layers, such as two consecutive convolutional layers in VGG, to form a block. This structure is similar to a residual block, however, since we are training a single-branch architecture instead of a real ResNet, the virtual shortcut connection is only used when the main path is pruned by DropPath during training. That is to say when the main path is not pruned, the virtual shortcut connection is removed so that we are still training a single-branch network. Correspondingly, the virtual shortcut connection is discarded during inference.

**Improved Shortcut Connection:** In the original ResNet (He et al., 2016), if one residual block's input shape is the same as its output shape, the shortcut connection as in Figure 2(c) is just an identity function, otherwise a $1 \times 1$ convolution layer of a stride larger than one, which may be followed by a normalization layer, is adopted in the shortcut connection to transform the input's shape to match the output's. In the latter case, the resolution of the feature maps is divided by the stride. For example, if the stride is 2, the top left entry in each $2 \times 2$ area of the input feature map is sampled, whereas the rest 3 entities of the same area are directly dropped.

This naive subsampling strategy will cause dramatic information loss when we use DropPath. Specifically, if DropPath prunes the main path as in Figure 2 (a), the shortcut connection will dominate the output of the residual block. In this regard, the naive subsampling strategy may corrupt or degrade the quality of the features, since it always picks a fixed entry of a grid. To tackle this issue, we replace the original shortcut connect with a $2 \times 2$ max pooling followed by a $1 \times 1$ convolutional layer with the stride of $1$. This improved structure will preserve the most important information after pooling instead of the one from a fixed entry. Figure 2 (c) and (d) show the comparison between the original and improved shortcut connections when the shapes of input and output are different.

## 3.2 KNOWLEDGE DISTILLATION

Given sufficient training data, large models usually perform better than small models due to their larger representation capability. Knowledge distillation aims to compress a well-trained large model (i.e., teacher model) into a smaller model (i.e., student model) without compromising too much performance. The basic idea behind knowledge distillation is to distill the knowledge from a teacher model into a student model by forcing the students predictions (or internal activations) to match those of the teacher (Beyer et al., 2022). Specifically, we can use Kullback-Leibler (KL) divergence with temperature $\mathcal{L}_{KL}$ (Hinton et al., 2015) to match the predictions of student and teacher models. Then, we can combine the KL divergence as the regularization term in addition to the classification loss. Mathematically, the overall loss is:

$$\mathcal{L}(\mathbf{y}_s, \mathbf{y}_t, y) = \alpha \cdot \tau^2 \cdot \mathcal{L}_{KL}(\mathbf{y}_s, \mathbf{y}_t) + (1 - \alpha) \cdot \mathcal{L}_{CE}(\mathbf{y}_s, y), \qquad (2)$$

where $\tau$ denotes the temperature factor, and $\alpha \in (0, 1)$ denotes the weight factor to balance the KL divergence $\mathcal{L}_{KL}$ and cross-entropy $\mathcal{L}_{CE}$. The output logits of the student model and teacher model are denoted by $\mathbf{y}_s$ and $\mathbf{y}_t$, respectively. $y$ denotes the target.

In our context, small models can perform better than large ones, since small models are used to construct distilled dataset. As a result, we adopt the small training network as the teacher model and the large test network as the student model. The computational overhead in knowledge distillation mainly arises from calculating $\mathbf{y}_t$. In this case, the computational overhead is negligible because evaluating on the small teacher network is more efficient than on the larger student network.

## 3.3 Training and Data Augmentation

Besides aforementioned methods, we use the following methods to further improve the performance.

**Periodical Learning Rate:** Because of the three-phase stepwise scheduler for the keep rate $p$, we expect the network to jump out of the current local minima, and tries to search for a better one when $p$ changes. Inspired by (Huang et al., 2017), we use a cosine annealing curve with warmup to adjust the learning rate, and we periodically reset it when $p$ changes. Formally, the learning rate is adjusted as shown in Eq. 3 of Appendix A.1.

**Better Optimizer:** Lion (Chen et al., 2023) is a gradient symbol-based optimizer. It has faster convergence speed and is capable of finding better local minima for ResNets. Thus, Lion is used as the default optimizer in our experiments.

**Stronger Augmentation:** The data augmentation strategy used in MTT (Cazenavette et al., 2022) samples a single augmentation operation from a pool to augment the input image. However, we observe that sampling more operations will better diversify the model's inputs and thus improve the performance, especially when IPC is small. For convenience, when sampling $k$ operations, we call this strategy $k$-fold augmentation. Empirically, we use 2-fold augmentation when IPC is 10 or 50 and 4-fold augmentation when IPC is 1. For the experiments about the impact of different augmentations, please refer to Appendix B.4.

## 4 Experiments

In this section, we evaluate our method on different dataset distillation algorithms, different numbers of instances per class (IPCs), different datasets and different network architectures. Our methods are shown effective in mitigating architecture overfitting and generic to improve the performance on limited real data. In addition, we conduct extensive ablation studies for analysis. Implementation details are referred to Appendix C.

### 4.1 Mitigate Architecture Overfitting in Dateset Distillation

We first evaluate our method on two representative dataset distillation (DD) algorithms, i.e., neural Feature Regression with Pooling (FRePo) (Zhou et al., 2022) and Matching Training Trajectories (MTT) (Cazenavette et al., 2022). FRePo proposes a neural feature kernel to solve a kernel ridge regression problem, and MTT focuses on matching the training trajectories on real data. Both of them show competitive performance. Furthermore, we test several ablations of our methods, and the settings of each ablation are elaborated in Table 1. Comparison with other baselines is shown in Appendix B.1.

Table 1: Experimental settings. *DP* denotes DropPath with three-phase keep rate, *KD* denotes knowledge distillation. Besides, the miscellaneous (Misc.) includes the methods in Section 3.3.

| Method | DP | KD | Misc. |
|---|---|---|---|
| Baseline | ✘ | ✘ | ✘ |
| w/o DP & KD | ✘ | ✘ | ✔ |
| w/o DP | ✘ | ✔ | ✔ |
| w/o KD | ✔ | ✘ | ✔ |
| Full | ✔ | ✔ | ✔ |

We comprehensively evaluate the performance of these methods under various settings, including different numbers of instances per class (IPC), different datasets and different architectures of the test networks. Table 2 demonstrates the results on CIFAR10, and the results on CIFAR100 and Tiny-ImageNet are reported in Appendix B.2. Note that, DropPath and knowledge distillation are not applicable when we use the same architecture for training and test networks, i.e., 3-layer CNN, because 1) it is too shallow for DropPath; 2) we will converge to the teacher model if we use the same architecture for the teacher and the student models. We can observe from these results that architecture overfitting is more severe in the case of small IPC and large architecture discrepancy, but both DropPath and knowledge distillation is capable of mitigating it. In addition, combining them can further improve the performance and overcome architecture overfitting in many cases. For instance, when evaluating our method on distilled images of MTT (CIFAR10, IPC=10), it contributes performance gains of $18.5\%$ and $35.7\%$ for ResNet18 and ResNet50, respectively. We are also interested in how much performance gap between training and test networks we can close. Surprisingly, when IPC=10 and 50, the test accuracies of most network architectures surpass that of the architec-

ture identical to the training network. Along with it, the gaps between different test networks, such as ResNet18 and ResNet50, are also narrowed down in most cases. Additionally, we observe that FRePo shows better cross-architecture generalization than MTT.

Table 2: Test accuracies of models trained on the distilled data of **CIFAR10** (Krizhevsky et al., 2009) with different IPCs. 3-layer CNN is the architecture used for data distillation and is the teacher model of knowledge distillation. The results in the bracket indicate the gaps from the baseline performance of 3-layer CNN. The results in bold are the best results among different settings. Note that DP and KD are not applicable for 3-layer CNN, so we do not have the test accuracy of 3-layer CNN in these settings.

| DD | IPC | Methods | 3-layer CNN | ResNet18 | AlexNet | VGG11 | ResNet50 |
|---|---|---|---|---|---|---|---|
| | | Baseline | 44.3 | 34.4 (-9.9) | 41.8 (-2.5) | 44.0 (-0.3) | 25.9 (-18.4) |
| | | w/o DP & KD | **44.8** (+0.5) | 35.6 (-8.7) | 47.4 (+3.1) | 41.5 (-2.8) | 30.3 (-14.0) |
| | 1 | w/o DP | - | 47.2 (+2.9) | 49.7 (+5.4) | 48.7 (+4.4) | 39.3 (-5.0) |
| | | w/o KD | - | 37.0 (-7.3) | 46.0 (+1.7) | 41.1 (-3.2) | 32.5 (-11.8) |
| | | **Full** | - | **49.3** (+5.0) | **50.7** (+6.4) | **48.8** (+4.5) | **41.5** (-2.8) |
| | | Baseline | 63.0 | 55.6 (-7.4) | 59.3 (-3.6) | 61.3 (-1.7) | 44.4 (-18.6) |
| | | w/o DP & KD | **64.7** (+1.7) | 61.0 (-2.0) | 62.3 (-0.7) | 62.4 (-0.6) | 54.7 (-8.3) |
| FRePo (Zhou et al., 2022) | 10 | w/o DP | - | 64.0 (+1.0) | 63.3 (+0.3) | 63.6 (+0.6) | 57.7 (-5.3) |
| | | w/o KD | - | 63.9 (+0.9) | 63.8 (+0.8) | 62.2 (-0.8) | 54.0 (-9.0) |
| | | **Full** | - | **66.6** (+3.6) | **64.8** (+1.8) | **65.4** (+2.4) | **62.4** (-0.6) |
| | | Baseline | 70.5 | 66.7 (-3.8) | 66.8 (-3.7) | 68.3 (-2.2) | 60.5 (-10.0) |
| | | w/o DP & KD | **72.4** (+1.9) | 73.0 (+2.5) | 71.0 (+0.5) | 70.9 (+0.4) | 71.2 (+0.7) |
| | 50 | w/o DP | - | 73.9 (+3.4) | 72.1 (+1.6) | 72.0 (+1.5) | 72.9 (+2.4) |
| | | w/o KD | - | 74.5 (+4.0) | 71.5 (+1.0) | 70.1 (-0.4) | 70.6 (+0.1) |
| | | **Full** | - | **74.5** (+4.0) | **73.2** (+2.7) | **72.8** (+2.3) | **73.2** (+2.7) |
| | | Baseline | **48.3** | 37.2 (-11.1) | 40.5 (-7.8) | 39.3 (-9.0) | 22.4 (-25.9) |
| | | w/o DP & KD | 46.8 (-1.5) | 36.9 (-11.4) | 43.2 (-5.1) | 36.7 (-11.6) | 24.7 (-23.6) |
| | 1 | w/o DP | - | 41.6 (-6.7) | 46.7 (-1.6) | 38.6 (-9.7) | 32.4 (-15.9) |
| | | w/o KD | - | 35.5 (-12.8) | 41.1 (-7.2) | 34.4 (-13.9) | 28.5 (-19.8) |
| | | **Full** | - | **47.2** (-1.1) | **47.3** (-1.0) | **44.1** (-4.2) | **43.0** (-5.3) |
| | | Baseline | 63.6 | 48.9 (-14.7) | 56.9 (-6.7) | 52.6 (-11.0) | 28.1 (-35.5) |
| | | w/o DP & KD | **65.0** (+1.4) | 51.3 (-12.3) | 60.7 (-2.9) | 56.0 (-7.6) | 39.8 (-23.8) |
| MTT (Cazenavette et al., 2022) | 10 | w/o DP | - | 61.4 (-2.2) | 52.7 (-10.9) | 48.8 (-14.8) | 49.9 (-13.7) |
| | | w/o KD | - | 60.7 (-2.9) | 59.2 (-4.4) | 57.6 (-6.0) | 47.5 (-16.1) |
| | | **Full** | - | **67.4** (+3.8) | **68.3** (+4.7) | **67.1** (+3.5) | **63.8** (+0.2) |
| | | Baseline | 70.2 | 62.3 (-7.9) | 67.5 (-2.7) | 63.0 (-7.2) | 53.1 (-17.1) |
| | | w/o DP & KD | **70.5** (+0.3) | 68.1 (-2.1) | 69.5 (-0.7) | 67.6 (-2.6) | 66.5 (-3.7) |
| | 50 | w/o DP | - | 66.9 (-3.3) | 63.8 (-6.4) | 61.2 (-9.0) | 66.8 (-3.4) |
| | | w/o KD | - | 69.8 (-0.4) | 67.2 (-3.0) | 69.0 (-1.2) | 65.0 (-5.2) |
| | | **Full** | - | **71.0** (+0.8) | **72.0** (+1.8) | **69.5** (-1.2) | **70.0** (-0.2) |

DropPath enables an implicit ensemble of the shallow subnetworks and thus mitigates architecture overfitting. However, each of these sub-networks may have sub-optimal performance. Knowledge distillation can address this issue by encouraging similar outputs between the teacher model and the sub-networks and thus further improves the performance. By contrast, the contribution of knowledge distillation could be marginal without DropPath due to the big difference in architecture (Mirzadeh et al., 2020). Empirically, combining DropPath with knowledge distillation not only achieves the best performance, but also greatly decreases the performance difference among different test network architectures.

Due to space limits, we report the standard deviations of performance in Table 8 of Appendix B.3. The results show that although the standard deviations increase when decreasing IPC, we can still see significant improvement by our methods.

## 4.2    IMPROVE THE PERFORMANCE OF TRAINING ON LIMITED REAL DATA

We discuss the performance of our methods when training on a limited amount of real data and compare it with the case of the distilled dataset. Our methods have shown effective to mitigate ar-

chitecture overfitting on the distilled dataset, we expect them to improve the performance on limited real training data as well. In this case, smaller models also tend to perform better than larger models because both can fit the training set perfectly but the latter suffers more from overfitting.

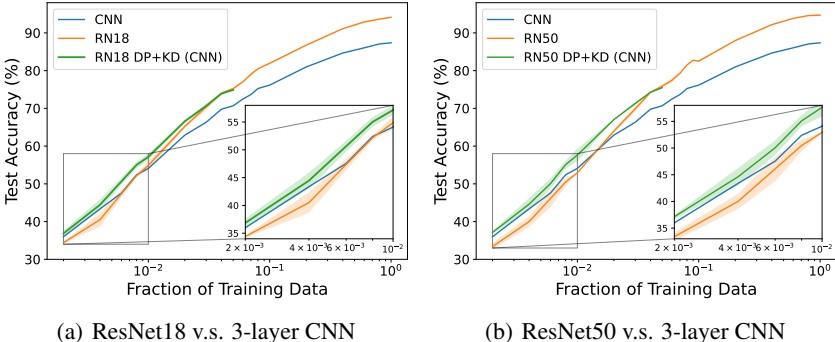

(a) ResNet18 v.s. 3-layer CNN        (b) ResNet50 v.s. 3-layer CNN

Figure 3: Test accuracies obtained from training on different fractions of CIFAR10, the shadow indicates the standard deviation. We compare the test accuracies **(a)** between ResNet18 (RN18) and 3-layer CNN (CNN), and **(b)** between ResNet50 (RN50) and CNN, respectively. The x-axis denotes the fraction of training data, *DP+KD* denotes that the network is trained with DropPath and knowledge distillation. Note that we run the experiments three times with different random seeds.

As illustrated in Figure 3, we train models on different fractions of CIFAR10 training set which are randomly sampled. The 3-layer CNN still serves as the teacher model when we use knowledge distillation. Since ResNet18 and ResNet50 exhibit the largest performance differences from the 3-layer CNN in the previous experiments, we only show the results of ResNet18 and ResNet50 here. ResNet18 and ResNet50 significantly outperform 3-layer CNN with enough training data, but they show worse generalization performance than CNN when the fraction is lower than 0.02, i.e., 1000 training instances. Under our methods, the performances of both ResNet18 and ResNet50 surpass that of 3-layer CNN even when the fraction is as small as 0.002, i.e., 100 training instances. However, the performance gain saturates when the fraction of training data reaches 0.05, which can be attributed to the unsatisfactory performance of the teacher model (blue line). Therefore, we do not bother to obtain the results with larger fractions as a result. Nevertheless, Figure 7 (b) in Appendix B.5 shows that when the current teacher does not contribute to performance gain anymore, a stronger teacher can further improve the performance. More results are discussed there.

Furthermore, we observe that the performance gap of training on limited real data is much smaller than that of training on distilled images. For instance, when the fraction of training data is 0.002, which is equivalent to IPC=10, the performance gap between 3-layer CNN and ResNet50 is 4.9% when they are trained on real images. However, when we train them on distilled images of FRePo, the performance gap increases to 18.6%. As for the distilled images generated by MTT, the gap is even larger, which reaches 35.5%. Meanwhile, training on a distilled dataset results in much better performance than training on real data of the same size, which makes it popular in downstream applications. Therefore, we focus on applying our method in the context of dataset distillation, in which the effectiveness of our method can be better revealed.

### 4.3 ABLATION STUDIES

We conduct extensive ablation studies here to validate the effectiveness of each component in our methods. In this subsection, we focus on the case of using 3-layer CNN as the training network, ResNet18 as the test network, setting IPC to 10 and generating the distilled dataset by FRePo. Note that the baseline performance of 3-layer CNN trained on the distilled data is 63.0%, its performance improves to 64.7% with better optimization and data augmentation.

**DropPath:** We first try different minimum keep rates in the three-phase scheduler introduced in Section 3.1. As illustrated in Figure 4 (a) and (c), a lower minimum keep rate and a longer period of decay induce better performance, but both of them make the training longer. To balance performance and efficiency, we set the minimum keep rate and period of decay to 0.5 and 500, respectively. Figure

4 (b) shows that different final keep rates do not significantly affect the performance. Moreover, we verify the effectiveness of the high keep rate in the final phase of training, and the improved shortcut connection (SC) introduced in Section 3.1. The results shown in Table 3 indicate that both of them contribute to the performance.

Table 3: Ablation studies about the high keep rate in the final phase of training and improved shortcut connection (SC). Note that if we do not adopt the improved SC, we use the original one instead.

| Final phase | Improved SC | Test Acc. |
|:---:|:---:|:---:|
| ✗ | ✗ | 65.2 |
| ✔ | ✗ | 65.6 |
| ✗ | ✔ | 65.9 |
| ✔ | ✔ | **66.6** |

Table 4: Ablation studies about optimization and data augmentation. If periodical learning rate (LR), Lion optimizer and stronger augmentation (Aug.) are not adopted, we replace them with cosine annealing learning rate, AdamW and 1-fold augmentation, respectively.

| Periodical LR | Lion | stronger Aug. | Test Acc. |
|:---:|:---:|:---:|:---:|
| ✗ | ✗ | ✗ | 61.6 |
| ✔ | ✗ | ✗ | 61.9 |
| ✔ | ✔ | ✗ | 64.8 |
| ✔ | ✔ | ✔ | **66.6** |

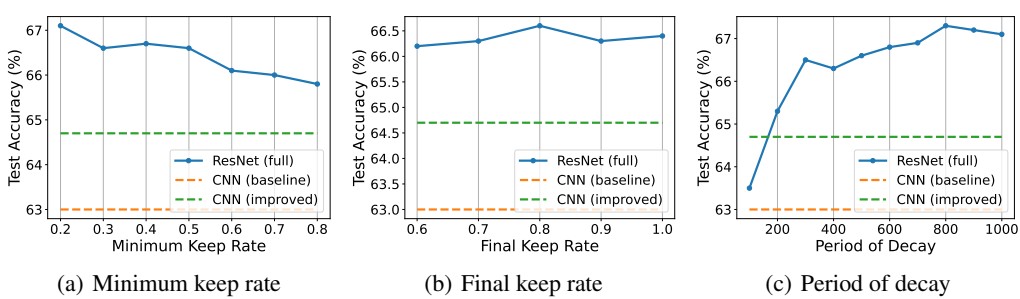

(a) Minimum keep rate      (b) Final keep rate      (c) Period of decay

Figure 4: Ablation studies on minimum keep rate, final keep rate and period of decay. **(a)** Test accuracies of different minimum keep rates. **(b)** Test accuracies of different keep rates at the final phase. **(c)** Test accuracies of different periods of decay. Regardless of the variation of hyperparameters, ResNet18 trained with our approach generally outperforms 3-layer CNN trained with baseline (orange dashed line) and that trained with better optimization and data augmentation (green dashed line).

**Knowledge Distillation:** We also test different hyperparameters of knowledge distillation (KD). As illustrated in Figure 8 (a) and (b) of Appendix B.6, when weight $\alpha$ and temperature $\tau$ are in the range of [0.5, 0.8] and [1, 10], respectively, the performance does not vary significantly. It indicates that our method is quite robust to different hyperparameter choices.

**Optimization and Data Augmentation:** In Table 4, we replace each of the optimization and data augmentation approaches with a baseline. The results indicate that each of these approaches improves performance. Among them, Lion optimizer contributes a performance improvement of 2.9%. Compared with adaptive optimizers, such as AdamW (Loshchilov & Hutter, 2017), Lion tends to converge to flatter minima, which results in better generalization performance (Keskar & Socher, 2017; Zhou et al., 2020). Figure 9 of Appendix B.7 further demonstrates that Lion finds flatter minima than AdamW from both quantitative and qualitative perspectives by numerical methods.

Note that the results of IPC=1 in Table 2 are obtained with 4-fold augmentation. For comparison, we also get the results with 2-fold augmentation (see in Table 9 of Appendix B.4).

## 5 CONCLUSION

This paper studies architecture overfitting when we train models on distilled datasets. We propose a series of approaches in both architecture designs and training schemes which can be adopted together to mitigate this issue. Our methods are efficient, generic and can improve the performance when training on a small real dataset directly. We believe that our work can help extend dataset distillation for applications in more real-world scenarios. Recognizing the existing disparity in performance between training on distilled data and the original training set, our future work will focus on exploring methods to further enhance performance.

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

# A SUPPLEMENTARY OF METHODS

## A.1 DROPPATH WITH THREE-PHASE KEEP RATE

The pseudo algorithm of DropPath with three-phase keep rate is shown in Alg. 1.

---

**Algorithm 1** DropPath with Three-Phase Keep Rate

---

**Input:** the data: $\mathbf{x}$; current epoch index: $i$; decaying factor: $0 < \gamma < 1$; minimum keep rate: $p_{min}$; final keep rate: $p_{final}$; period of decay: $T$; warmup period: $W$; stabilization epoch: $S$.

1: **if** $i < W$ **then**
2:     $p \leftarrow 1$
3: **else if** $i < S$ **then**               ▷ ceil function returns the smallest integer bigger than the input
4:     $p \leftarrow \mathtt{max}(p_{min}, 1 - \gamma \cdot \mathtt{ceil}((i - W)/T))$
5: **else**
6:     $p \leftarrow p_{final}$
7: **end if**
8: **if** is training **then**               ▷ sampled from Bernoulli distribution
9:     $m \leftarrow \mathtt{Bernoulli}(p)$
10:     $\mathbf{y} \leftarrow \frac{m}{p} \cdot \mathbf{x}$
11: **else**
12:     $\mathbf{y} \leftarrow \mathbf{x}$
13: **end if**
    **Output:** $\mathbf{y}$

---

Unless specified, we set $\gamma = 0.1$, $p_{min} = 0.5$, $p_{final} = 0.8$, $T = 500$, $W = 500$, $S = 3000$ in the experiments. The corresponding curve of the dynamic keep rate is shown in Figure 5.

The learning rate $lr$ at epoch $i$ is defined as

$$lr_i = \begin{cases} \lambda_i \cdot \frac{\mathrm{mod}(i,t)}{T_{warm}} \cdot lr_{max}, & \text{if } \mathrm{mod}(i,t) \leq T_{warm}, \\ 0.5\lambda_i(1 + \cos(\pi \frac{\mathrm{mod}(i,t) - T_{warm}}{T_{max} - T_{warm}})) \cdot lr_{max}, & \text{else,} \end{cases} \tag{3}$$

where $t = T$ when $i < S$, otherwise $t = S$. $\lambda_i = \lambda^{\lfloor \mathtt{min}(i,S)/T \rfloor}$ where $\lambda$ is a base decaying factor, and $\lfloor \cdot \rfloor$ denotes the floor function. $lr_{max}$ denotes the maximum learning rate, $\mathrm{mod}(x, y)$ denotes the remainder of $x/y$, $T$ is the decay period of the keep rate $p$ of DropPath, $S$ is the stabilization epoch. The maximum iterations of the cosine annealing function and the number of warmup epochs are denoted by $T_{max}$ and $T_{warm}$, respectively. Figure 6 shows how the learning rate changes.

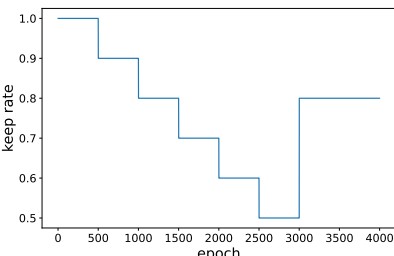

Figure 5: Scheduler of three-phase keep rate.

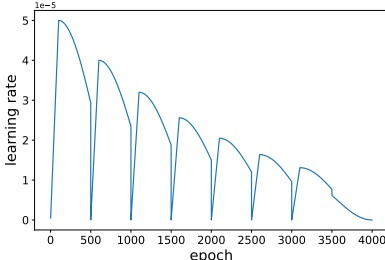

Figure 6: Curve of periodical learning rate.

## A.2 EFFECT OF SCALING FACTOR $1/p$ IN DROPPATH

Eq. 1 shows that $\mathtt{DropPath}(\mathbf{x}) = \frac{m}{p} \cdot \mathbf{x}$,     $m = \mathtt{Bernoulli}(p)$, where $p \in [0, 1]$ denotes the keep rate, $m = \mathtt{Bernoulli}(p) \in \{0, 1\}$ outputs 1 with probability $p$ and 0 with probability $1 - p$. We consider the module output $\mathbf{y} = \mathtt{DropPath}(\mathbf{x})$ in the training phase, then the expectation of $\mathbf{y}$ given

$x$ is $\mathbb{E}(y) = p \cdot \frac{1}{p} \cdot x + (1-p) \cdot \frac{0}{p} \cdot x = x$. In the test phase, DropPath is disabled, so the module output is simply $x$ and consistent with the expectation in the training phase. If there is no scaling factor $1/p$ in Eq. 1 and $p < 1$, the expectation of the module outputs in the training and test phases will be different, which leads to performance degradation.

## B    ADDITIONAL EXPERIMENTS

### B.1    COMPARISON WITH OTHER BASELINES

Table 5: Comparison between our method and other baselines. Note that p in DropPath and Dropout denotes the keep rate and alpha in MixUp and CutMix is the parameters $\alpha$ and $\beta$ in beta distribution, where $\alpha$ and $\beta$ are the same.

| Method | Test Accuracy |
|---|---|
| Baseline | 55.6 |
| DropPath (p=0.5) | 46.2 |
| DropOut (p=0.5) | 44.3 |
| MixUp (alpha=0.5) | 57.6 |
| CutMix (alpha=0.5) | 56.9 |
| **Ours** | **66.2** |

We further compare our method with some regularization methods on architectures: DropOut (Srivastava et al., 2014) and DropPath (Larsson et al., 2016), and data augmentation methods: MixUp (Zhang et al., 2017) and CutMix (Yun et al., 2019). We evaluate their performance under the setting of using 3-layer CNN as the training network, ResNet18 as the test network, setting IPC to 10 and generating the distilled dataset by FRePo. The result is reported in Table 5. We observe that Drop-Path and DropOut with a constant keep rate deteriorate the performance; MixUp and CutMix only contribute to marginal performance improvement, which further demonstrates the effectiveness of our method.

### B.2    RESULTS ON MORE DATASETS

The results on CIFAR100 and Tiny-ImageNet are reported in Table 6 and 7, respectively. The observations on CIFAR10 are analogous to those on CIFAR100 and Tiny-ImageNet, which indicates that our method is effective on different datasets.

### B.3    STANDARD DEVIATION OF RESULTS

To better validate the effectiveness of our method, we report the standard deviations of test accuracies of CIFAR10 (FRePo) in Table 8. We calculate these standard deviations by running the experiments three times with different random seeds. It can be observed that the standard deviation generally increases as IPC decreases. The reason could be that when IPC gets smaller, there are more solutions that make the training error zero, so the performance of training becomes more sensitive to initialization. Despite that, we can still see significant improvement by our methods.

### B.4    THE IMPACT OF AUGMENTATION WHEN IPC=1

In Table 2, we use 4-fold augmentation when IPC = 1 and 2-fold augmentation otherwise. To comprehensively analyze the effect of a stronger data augmentation, we demonstrate the results of using 2-fold augmentation in the case of distilled data of CIFAR10 (FRePo, IPC = 1) in Table 9. Compared with Table 2, the test accuracies of *w/o DP & KD* and *w/o & KD* in Table 9 are higher, but those of *w/o DP* and *Full* are lower. Especially for ResNet50, the performance of *Full* increases by 7.7% with 4-fold augmentation. This indicates that stronger augmentation is necessary when using knowledge distillation when there are extremely limited data, and when the architecture difference between the training and test networks is big. Moreover, we observe that the contribution of 4-fold augmentation is marginal under a larger IPC, so we adopt 4-fold augmentation only when IPC=1.

Table 6: Test accuracies of models trained on the distilled data of **CIFAR100** (Krizhevsky et al., 2009) with different IPCs. 3-layer CNN is the architecture used for data distillation and is the teacher model of knowledge distillation. The results in the bracket indicate the gaps from the baseline performance of 3-layer CNN. The results in bold are the best results among different methods.

| DD | IPC | Methods | 3-layer CNN | ResNet18 | AlexNet | VGG11 | ResNet50 |
|---|---|---|---|---|---|---|---|
| FRePo (Zhou et al., 2022) | 1 | Baseline | **26.2** | 18.7 (-7.5) | 22.9 (-3.3) | 22.6 (-3.6) | 13.5 (-12.7) |
| | | w/o DP & KD | 26.1 (-0.1) | 16.0 (-10.2) | 22.3 (-3.9) | 18.4 (-7.8) | 14.5 (-11.7) |
| | | w/o DP | - | 21.3 (-4.9) | 23.9 (-2.3) | 21.8 (-4.4) | 18.2 (-8.0) |
| | | w/o KD | - | 17.1 (-9.1) | 22.1 (-4.1) | 17.9 (-8.3) | 14.3 (-11.9) |
| | | **Full** | - | **24.4** (-1.8) | **25.3** (-0.9) | **24.0** (-2.2) | **23.7** (-2.5) |
| | 10 | Baseline | 34.4 | 32.1 (-2.3) | 33.1 (-1.3) | 34.1 (-0.3) | 28.1 (-6.3) |
| | | w/o DP & KD | 40.2 (+5.8) | 35.3 (+0.9) | 37.9 (+3.5) | 37.2 (+2.8) | 33.7 (-0.7) |
| | | w/o DP | - | 39.4 (+5.0) | 39.2 (-4.8) | 38.9 (+4.5) | 38.5 (+4.1) |
| | | w/o KD | - | 34.8 (+0.4) | 38.5 (+4.1) | 36.6 (+2.2) | 35.0 (+0.6) |
| | | **Full** | - | **40.6** (+6.2) | **39.9** (+5.5) | **39.4** (+5.0) | **40.1** (+5.7) |
| | 50 | Baseline | 42.1 | 46.7 (+4.6) | 45.5 (+3.4) | 45.5 (+3.4) | 45.8 (+3.7) |
| | | w/o DP & KD | 46.2 (+4.1) | 46.8 (+4.7) | 46.1 (+4.0) | 45.5 (+3.4) | 46.9 (+4.8) |
| | | w/o DP | - | 48.3 (+6.2) | 44.6 (+2.5) | 45.8 (+3.7) | 48.7 (+6.6) |
| | | w/o KD | - | 47.2 (+5.1) | **47.0** (+4.9) | 45.0 (+2.9) | 46.1 (+4.0) |
| | | **Full** | - | **48.5** (+6.4) | 46.6 (+4.5) | **46.7** (+4.6) | **49.1** (+7.0) |
| MTT (Cazenavette et al., 2022) | 1 | Baseline | 24.4 | 14.3 (-10.1) | 17.0 (-7.4) | 15.6 (-8.8) | 4.6 (-19.8) |
| | | w/o DP & KD | **25.0** (+0.6) | 12.5 (-11.9) | 20.6 (-3.8) | 8.2 (-16.2) | 6.0 (-18.4) |
| | | w/o DP | - | 13.3 (-11.1) | 24.4 (+0.0) | 10.2 (-14.2) | 8.5 (-15.9) |
| | | w/o KD | - | 13.6 (-10.8) | 19.7 (-4.7) | 12.4 (-12.0) | 9.3 (-15.1) |
| | | **Full** | - | **24.9** (+0.5) | **25.8** (+1.4) | **22.1** (-2.3) | **24.6** (+0.2) |
| | 10 | Baseline | 38.4 | 32.9 (-5.5) | 33.7 (-4.7) | 28.8 (-9.6) | 22.5 (-15.9) |
| | | w/o DP & KD | **38.5** (+0.1) | 32.7 (-5.7) | 36.0 (-2.4) | 33.9 (-4.5) | 30.6 (-7.8) |
| | | w/o DP | - | 35.0 (-3.4) | 38.2 (-0.2) | 35.5 (-2.9) | 34.2 (-4.2) |
| | | w/o KD | - | 34.6 (-3.8) | 34.9 (-3.5) | 33.2 (-5.2) | 32.9 (-5.5) |
| | | **Full** | - | **38.4** (+0.0) | **39.9** (+1.5) | **36.4** (-2.0) | **38.5** (+0.1) |
| | 50 | Baseline | 44.5 | 43.1 (-1.4) | 41.4 (-3.1) | 39.3 (-5.2) | 38.7 (-5.8) |
| | | w/o DP & KD | **46.0** (+1.5) | 46.2 (+1.7) | 46.1 (+1.6) | 44.5 (+0.0) | 45.5 (+1.0) |
| | | w/o DP | - | 47.2 (+2.7) | 47.1 (+2.6) | 45.1 (+0.6) | 47.2 (+2.7) |
| | | w/o KD | - | 46.9 (+2.4) | 45.7 (+1.2) | 43.4 (-1.1) | 46.8 (+2.3) |
| | | **Full** | - | **48.9** (+4.4) | **47.6** (+3.1) | **45.1** (+0.6) | **49.4** (+4.9) |

Table 7: Test accuracies of models trained on the distilled data of **Tiny-ImageNet** (Deng et al., 2009) with different IPCs. 3-layer CNN is the architecture used for data distillation and is the teacher model of knowledge distillation. The results in the bracket indicate the gaps from the baseline performance of 3-layer CNN. The results in bold are the best results among different methods. Due to computational complexity, the results of IPC=50 are not included.

| DD | IPC | Methods | CNN | ResNet18 | AlexNet | VGG11 | ResNet50 |
|----|-----|---------|-----|----------|---------|-------|----------|
| FRePo | 1 | Baseline | 16.6 | 15.6 (-1.0) | 16.5 (-0.1) | 16.6 (+0.0) | 13.4 (-3.2) |
| | | w/o DP & KD | **17.7** (+1.1) | 12.3 (-4.3) | 13.7 (-2.9) | 14.1 (-2.5) | 12.8 (-3.8) |
| | | w/o DP | - | 15.8 (-0.8) | 16.6 (+0.0) | 16.4 (-0.2) | 16.6 (+0.0) |
| | | w/o KD | - | 12.5 (-4.1) | 14.9 (-1.7) | 13.6 (-3.0) | 11.9 (-4.7) |
| | | **Full** | - | **18.9** (+2.3) | **18.5** (+1.9) | **18.3** (1.7) | **19.1** (+2.5) |
| | 10 | Baseline | **24.9** | 24.2 (-0.7) | 24.8 (-0.1) | 25.2 (+0.3) | 24.9 (+0.0) |
| | | w/o DP & KD | 23.0 (-1.9) | 21.7 (-3.2) | 23.8 (-1.1) | 24.2 (-0.7) | 23.1 (-1.8) |
| | | w/o DP | - | 25.4 (+0.5) | **25.2** (+0.3) | 26.4 (+1.5) | 26.9 (+2.0) |
| | | w/o KD | - | 21.5 (-3.4) | 22.4 (-2.5) | 24.0 (-0.9) | 21.6 (-3.3) |
| | | **Full** | - | **26.8** (+1.9) | 24.9 (+0.0) | **26.6** (+1.7) | **27.3** (+2.4) |
| MTT | 1 | Baseline | 8.8 | 6.2 (-2.6) | 6.7 (-2.1) | 7.3 (-1.5) | 2.7 (-6.1) |
| | | w/o DP & KD | **9.6** (+0.8) | 6.1 (-2.7) | 8.4 (-0.4) | 7.2 (-1.6) | 3.2 (-5.6) |
| | | w/o DP | - | 6.5 (-2.3) | 9.1 (+0.3) | 7.9 (-0.9) | 3.6 (-5.2) |
| | | w/o KD | - | 6.7 (-2.1) | 8.1 (-0.7) | 6.8 (-2.0) | 4.0 (-4.8) |
| | | **Full** | - | **8.1** (-0.7) | **9.2** (+0.4) | **8.2** (-0.6) | **8.2** (-0.6) |
| | 10 | Baseline | 19.3 | 17.2 (-2.1) | 14.3 (-5.0) | 15.1 (-4.2) | 11.2 (-8.1) |
| | | w/o DP & KD | **20.1** (+0.8) | 16.6 (-2.7) | 18.7 (-0.6) | 16.2 (-3.1) | 15.2 (-4.1) |
| | | w/o DP | - | 17.3 (-2.0) | 21.2 (+1.9) | 19.9 (+0.6) | 18.7 (-0.6) |
| | | w/o KD | - | 19.0 (-0.3) | 17.7 (-1.6) | 15.2 (-4.1) | 17.7 (-1.6) |
| | | **Full** | - | **22.6** (+3.3) | **21.6** (+2.3) | **20.5** (+1.2) | **21.8** (+2.5) |

Table 8: The average test accuracies of models trained on the distilled data of **CIFAR10** (Krizhevsky et al., 2009) with different IPCs. The number after ± denotes the standard deviation. These results are obtained through three repetitive experiments with different random seeds. 3-layer CNN is the architecture used in distillation and is the teacher model of knowledge distillation.

| DD | IPC | Methods | ResNet18 | AlexNet | VGG11 | ResNet50 |
|----|-----|---------|----------|---------|-------|----------|
| FRePo | 1 | w/o DP & KD | 35.6 ±2.5 | 47.4 ±0.9 | 41.5 ±1.1 | 30.3 ±1.9 |
| | | w/o DP | 47.2 ±0.5 | 49.7 ±0.7 | 48.7 ±0.6 | 39.3 ±1.4 |
| | | w/o KD | 37.0 ±1.0 | 46.0 ±0.6 | 41.1 ±1.3 | 32.5 ±1.4 |
| | | Full | 49.3 ±0.6 | 50.7 ±0.1 | 48.8 ±0.4 | 41.5 ±1.0 |
| | 10 | Full | 66.2 ±0.5 | 64.8 ±0.9 | 65.4 ±0.2 | 62.4 ±0.9 |
| | 50 | Full | 74.5 ±0.1 | 73.2 ±0.3 | 72.8 ±0.0 | 73.2 ±0.2 |

## B.5 ADDITIONAL RESULTS FOR SEC. 4.2

Figure 7 (a) indicates that our method is also effective for VGG11. Furthermore, Figure 7 (b) illustrates that when the fraction of training data is larger than 0.04, 3-layer CNN can no longer provide performance gains for ResNet50. Nonetheless, if we adopt ResNet18 as the teacher model at this point, the performance of ResNet50 can be further improved.

## B.6 ABLATION STUDY ON HYPERPARAMETERS OF KNOWLEDGE DISTILLATION

From Figure 8 (a) and (b), we can observe that when weight $\alpha$ and temperature $\tau$ are in the range of [0.5, 0.8] and [1, 10], respectively, the performance does not vary significantly. It indicates that our method is quite robust to different hyperparameter choices.

Table 9: Test accuracies of models trained on the distilled data of CIFAR10 (FRePo, IPC=1). However, 2-fold augmentation is adopted here. Except that, the other settings are the same as Table 2.

| IPC | Methods | 3-layer CNN | ResNet18 | AlexNet | VGG11 | ResNet50 |
|---|---|---|---|---|---|---|
| | Baseline | 44.3 | 34.4 (-9.9) | 41.8 (-2.5) | 44.0 (-0.3) | 25.9 (-18.4) |
| | w/o DP & KD | **44.8** (+0.5) | 41.2 (-3.1) | 45.4 (+1.1) | 45.9 (+1.6) | 32.8 (-11.5) |
| 1 | w/o DP | - | 41.0 (-3.3) | 44.5 (+0.2) | **47.0** (+2.7) | 30.0 (-14.3) |
| | w/o KD | - | 39.4 (-4.9) | 47.1 (+2.8) | 38.9 (-5.4) | 31.0 (-13.3) |
| | **Full** | - | **45.5** (+1.2) | **47.8** (+3.5) | 46.7 (+2.4) | **33.8** (-10.5) |

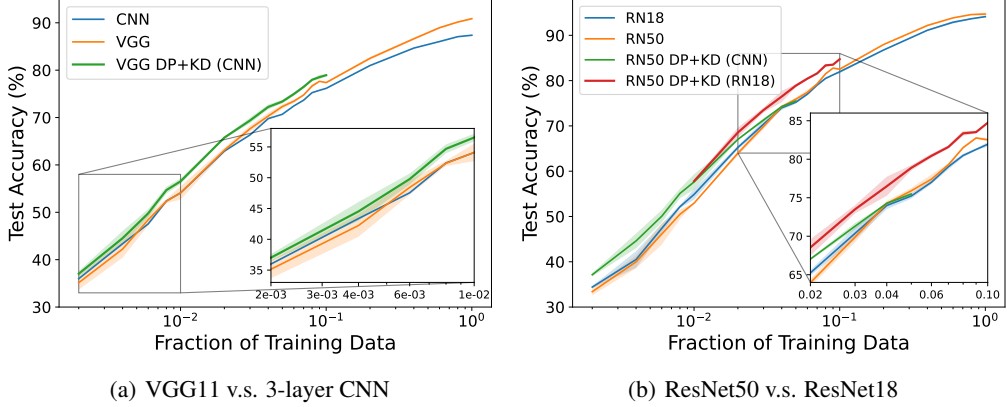

(a) VGG11 v.s. 3-layer CNN      (b) ResNet50 v.s. ResNet18

Figure 7: Test accuracies obtained from training on different fractions of CIFAR10, the shadow indicates the standard deviation. We compare the test accuracies **(a)** between VGG11 and 3-layer CNN (CNN), and **(b)** between ResNet50 (RN50) and ResNet18, respectively. The x-axis denotes the fraction of training data, *DP+KD* denotes that the model is trained with DropPath with three-phase keep rate and knowledge distillation, where the teacher architecture used in KD is shown in the bracket.

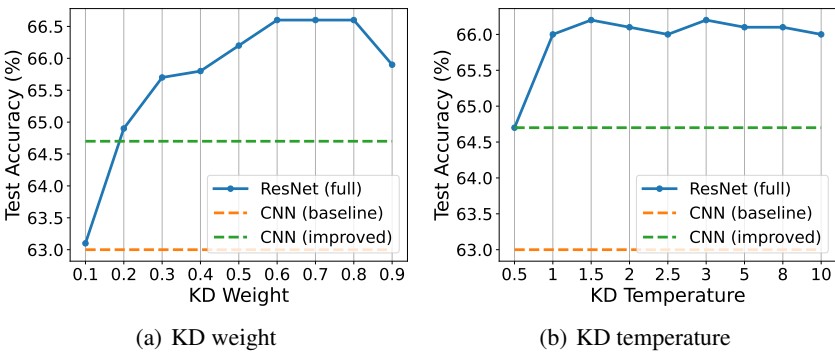

(a) KD weight      (b) KD temperature

Figure 8: Ablation studies on weight and temperature of knowledge distillation (KD). **(a)** Test accuracies of different KD weights. **(b)** Test accuracies of different KD temperatures. Regardless of the variation of hyperparameters, ResNet18 trained with our approach generally outperforms 3-layer CNN trained with baseline (orange dashed line) and that trained with better optimization and data augmentation (green dashed line).

## B.7 LION FINDS FLATTER MINIMA

To validate the claim in Sec. 4.3 that Lion finds flatter minima than AdamW, we analyze the Hessian spectrum of models trained with different optimizers. It is known that the curvature in the neighborhood of model parameters is dominated by the top eigenvalues of the Hessian matrix $\nabla^2 \mathcal{L}_{CE}(\theta)$, where $\mathcal{L}_{CE}(\theta)$ denotes the cross-entropy loss w.r.t model parameters $\theta$. In the implementation, we use the power iteration method as in (Yao et al., 2018; Liu et al., 2020) to iteratively estimate the top 20 eigenvalues and the corresponding eigenvectors of the Hessian matrix.

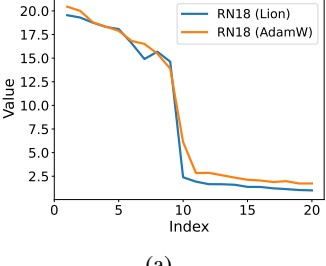 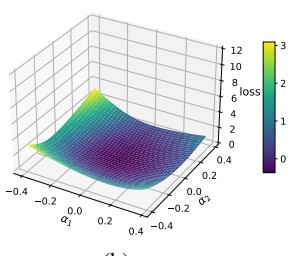 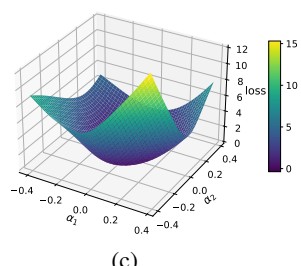

(a)            (b)            (c)

Figure 9: Additional results for Sec. 4.3. **(a)** Top 20 eigenvalues of Hessian matrix for ResNet18 trained with different optimizers, i.e., Lion and AdamW. **(b)** Loss landscape $\mathcal{L}_{CE}(\theta + \alpha_1 \mathbf{v}_1 + \alpha_2 \mathbf{v}_2)$ of ResNet18 around the minima found by Lion, where $\mathbf{v}_1$ and $\mathbf{v}_2$ are the eigenvectors corresponding to the top two eigenvalues of Hessian matrices, respectively. **(c)** The same loss landscape of ResNet18 around the minima found by AdamW. Note that the training data is 100 (IPC=10) distilled images of CIFAR10 by FRePo. ResNet18 is trained with DropPath and knowledge distillation. 3-layer CNN serves as the teacher model of knowledge distillation.

As illustrated in Figure 9 (a), the eigenvalues of the Hessian matrix for the ResNet18 trained with Lion are smaller than those for the ResNet18 trained with AdamW, which quantitatively indicates that the neighborhood of the minima found by Lion has smaller curvature. Furthermore, Figure 9 (b) and (c) qualitatively shows that Lion finds flatter minima than AdamW.

## C  IMPLEMENTATION DETAILS

**Datasets:** The training sets in the experiments are the distilled datasets of CIFAR10, CIFAR100 (Krizhevsky et al., 2009) and Tiny-ImageNet (Deng et al., 2009), but the test sets are their respective original test sets. To better validate the effectiveness of our method, we use the distilled images synthesized by different dataset distillation algorithms, e.g., neural Feature Regression with Pooling (FRePo) (Zhou et al., 2022) and Matching Training Trajectories (MTT) (Cazenavette et al., 2022). In addition, we evaluate the performance of our method in different numbers of instances per class (IPC), e.g., 1, 10 and 50. Note that MTT does not provide the final trainable learning rates in the released checkpoints, we adopt the reported initial learning rates in our baselines.

**Models:** The networks used to synthesize the distilled images (training networks) in FRePo and MTT are 3-layer CNN. Consistent with the hyperparameters reported in the paper, the output channels of hidden layers of the network used in FRePo are 128, 256 and 512, respectively. However, in MTT, all the output channels of hidden layers are set to 128. ResNet18, ResNet50 (He et al., 2016), AlexNet (Krizhevsky et al., 2012) and VGG11 (Simonyan & Zisserman, 2014) are adopted in the evaluation, they are thereby called test networks. The hyperparameters of networks are the same as those set in (Cazenavette et al., 2022). Note that when training networks on distilled images of FRePo and MTT, batch and instance normalization layers are adopted in networks following the settings of (Zhou et al., 2022; Cazenavette et al., 2022), respectively.

**DropPath:** As shown in Algorithm 1, the decaying factor of keep rate $\gamma = 0.1$, minimum keep rate $p_{min} = 0.5$, final keep rate $p_{final} = 0.8$, period of decay $T = 500$, warmup period $W = 50$, stabilization epoch $S = (1 + p_{min}/\gamma) \times T = 3000$. The total epochs $N$ is set to $S + 2 \times T = 4000$. In the improved shortcut, the pooling area depends on the stride of $1 \times 1$ convolutional layer in the original one. e.g., if the stride of $1 \times 1$ convolutional layer in the original shortcut is 2, we use a $2 \times 2$ max pooling.

**Knowledge distillation:** As shown in Eq. 2, the temperature factor $\tau = 1.5$, and the weight factor $\alpha = 0.5$. If not specifically indicated, the default teacher model is the 3-layer CNN. Note that the teacher model is trained on the same data set as the student model.

**Periodical learning rate:** In Eq. 3, the maximum learning rate $lr_{max} = 5 \times 10^{-5}$, the base decaying factor for learning rate $\lambda = 0.8$. The period of the cosine function $T_{max}$ and the number of warmup epochs $T_{warm}$ are 1000 and 50, respectively.

**Optimizer:** Lion (Chen et al., 2023) is adopted in our method, where weight decay $\lambda_{wd} = 5 \times 10^{-3}$, coefficient $\beta_1 = 0.95$, and $\beta_2 = 0.98$.

**Augmentation:** There are color jittering, cropping, cutout, flipping, scaling, and rotating in the augmentation pool, we sample more operations instead of just one as in (Cazenavette et al., 2022).

**Training:** For CIFAR10, the batch sizes for different IPCs are 10 (IPC=1), 100 (IPC=10) and 128 (IPC=50), respectively. For CIFAR100, the batch sizes are 100 (IPC=1), 256 (IPC=10) and 256 (IPC=50), respectively. Cross-entropy is adopted as the loss function in our experiments. Since the labels of images are learnable in FRePo, we divide them with a temperature factor $t = 0.3$ for CIFAR10, $0.04$ for CIFAR100, and and $0.02$ for Tiny-ImageNet.

