# OpenReview forum: "Towards Mitigating Architecture Overfitting in Dataset Distillation"
_ICLR.cc/2024/Conference — Submitted to ICLR 2024_

### Official Review · Reviewer_w1VW · 2023-10-22

**Soundness:** 3 good
**Presentation:** 3 good
**Contribution:** 3 good
**Rating:** 5
**Confidence:** 4

**Summary:**

This paper proposed a series of methods to address the cross-architecture generalization problem of dataset distillation. In detail, combining the DropPath and knowledge distillation, this paper proposed two adapted methods to use a new model design and loss objective to alleviate the overfitting problem. Besides, other tricks like LR and augmentation policy are also used. In experiments, the proposed method performed decently on the cross-architecture tests.

**Strengths:**

+ The proposed method achieved good results on the claimed cross-architecture test.

+ The presentation is easy to follow. The code is provided.

+ The experiments on multiple datasets, models, and settings provide a solid validation for the contribution.

**Weaknesses:**

- The results are impressive, however, the method contributions seem relatively marginal. Though the adapted method absorbed from previous works is non-trivial, the discussion lacks enough insight but is empirical.

- The three-phase Keep Rate looks quite complex for tuning. How is the tuning complexity and robustness if we use the proposed method for many different models?

- Though the bag of methods works well, the whole paper gives the readers a feeling of separation.

- typo: in the abs, performance across different network architectures {

**Questions:**

1. There were only discussions on the residual architecture, why? There are also other multi-branch architectures.

2. "As a result, we can also expect DropPath to mitigate the architecture overfitting issue in dataset distillation. " --- any more detailed analysis?

3. "Architecture overfitting arises from deeper test networks" --- any citation or discussion?

---

> ### Author Response · Authors · 2023-11-20
>
> Dear reviewer w1VW,
>
> Thanks for your constructive suggestions and insightful comments. In response to your questions, we offer the following point-to-point answers.
>
> For the problems mentioned in Weaknesses:
>
> 1. \>\>\> The results are impressive, however, the method contributions seem relatively marginal. Though the adapted method absorbed from previous works is non-trivial, the discussion lacks enough insight but is empirical.
>
>     **Reply:** Although our design is motivated by some existing techniques such as DropPath and knowledge distillation, we adapt them in the context of dataset distillation. For DropPath, we make it applicable to single-branch networks and propose improved shortcut connection and three-phase keep rate to boost the performance. For knowledge distillation, we pioneer the use of small networks as teacher models, which is different from current common practices. In the experiments, our method significantly mitigates the architecture overfitting in dataset distillation and improves the generalization of large networks on small datasets. Section 3 comprehensively demonstrates the motivation of our proposed methods. More fundamental analysis of DropPath and knowledge distillation were well analyzed in many works [1 - 3].
>
> 2. \>\>\> The three-phase Keep Rate looks quite complex for tuning. How is the tuning complexity and robustness if we use the proposed method for many different models?
>
>     **Reply:** In the ablation study, we evaluate the performance under different hyperparameters (HPs). **Moreover, additional ablation studies on HPs of three-phase keep rate were added**. The result is reported below and in Figure 4 (b) and (c). Compared to the significant performance improvement brought by our method, if we choose different HPs in an **appropriate but wide** range, the performance fluctuation caused by different HPs is negligible. For example, when the final keep rate ranges from 0.6 to 1.0, KD weight ranges from 0.3 to 0.9 and KD temperature ranges from 1 to 10, the performance fluctuation is with 1 percentage point. In addition, although lower minimum keep rates and longer periods of decay induce better performance, they also mean a longer training phase. We set them to 0.5 and 500 to balance the efficiency and performance, respectively. Despite that, the performance fluctuation caused by these two HPs is still insignificant. Therefore, we can demonstrate that our training method is robust to different HPs. In addition, the selected HPs also perform well in different settings, including different distilled data, different IPCs and different test networks.
>
> 3. \>\>\> Though the bag of methods works well, the whole paper gives the readers a feeling of separation.
>
>     **Reply:** The main contribution of our work is proposing a bag of methods to mitigate architecture overfitting in dataset distillation. These methods perform like regularization from different perspectives, such as **network architecture** (i.e. DropPath with three-phase keep rate), and **training scheme** (i.e. knowledge distillation, data augmentation and optimizer). Instead of directly using these methods, we adapt them to our scenario to further improve the performance. In the experiments, we demonstrate that our method not only mitigates architecture overfitting in dataset distillation but also improves the generalization of large models on small datasets. We believe our work provides practitioners with a toolbox and some intuition to solve related problems.

---

> > ### Author Response · Authors · 2023-11-20
> > **Continue**
> >
> > For the problems mentioned in Questions:
> > 1. \>\>\> There were only discussions on the residual architecture, why? There are also other multi-branch architectures.
> >
> >     **Reply:** For the architectures evaluated in the experiments, we follow the setting of previous works [4, 5]. The multi-branch networks adopted in the evaluation are ResNets there.
> >
> > 2. \>\>\> "As a result, we can also expect DropPath to mitigate the architecture overfitting issue in dataset distillation. " --- any more detailed analysis?
> >
> >     **Reply:** In dataset distillation, the test accuracy of the test network trained on the distilled dataset degrades significantly when it has a different network architecture from the training network, especially when the test network is deeper than the training one. DropPath can reduce the effective depth of networks during training by stochastically removing paths, which is equivalent to training a shallow sub-network at each iteration. Through minimizing the loss, we expect the performance of each shallow sub-network to be similar to that of the training network. Similar to Dropout, we implicitly learn an ensemble of sub-networks by DropPath. During the inference phase, since each shallow sub-network can work well, the ensemble of them will be even better.
> >
> > 3. \>\>\> "Architecture overfitting arises from deeper test networks" --- any citation or discussion?
> >
> >     **Reply:** In previous works about dataset distillation [4, 5], the architecture overfitting is also observed and becomes more severe when the depth of networks increases (from 3-layer CNN to ResNet). In addition, in a more general scenario, severe overfitting also occurs when we train a large and deep model on tiny real datasets.
> >
> >
> > ***Table 1: Ablation study on the keep rate in the final phase.***
> > |  Final Keep Rate   | 0.6 | 0.7 | 0.8 | 0.9 | 1.0 |
> > |  ----  | ----  | ----  | ----  | ----  | ----  |
> > | Test Accuracy  | 66.2 | 66.3 | **66.6** | 66.3 | 66.4 |
> >
> >
> > ***Table 2: Ablation study on the period of decay.***
> > |  Period of Decay   | 100 | 200 | 300 | 400 | 500 | 600 | 700 | 800 | 900 | 1000|
> > |  ----  | ----  | ----  | ----  | ----  | ----  | ----  | ----  | ----  | ----  | ----  |
> > | Test Accuracy  | 63.5 | 65.3 | 66.5 | 66.3 | 66.6 | 66.8 | 66.9 | **67.3** | 67.2 | 67.1 |
> >
> >
> > > [1] Hahn, Sangchul, and Heeyoul Choi. "Understanding dropout as an optimization trick." Neurocomputing 398 (2020): 64-70.
> >
> > > [2] Wang, Hongjun et al. “CamDrop: A New Explanation of Dropout and A Guided Regularization Method for Deep Neural Networks.” Proceedings of the 28th ACM International Conference on Information and Knowledge Management (2019): n. pag.
> >
> > > [3] Phuong, Mary, and Christoph Lampert. "Towards understanding knowledge distillation." International conference on machine learning. PMLR, 2019.
> >
> > > [4] George Cazenavette, Tongzhou Wang, Antonio Torralba, Alexei A Efros, and Jun-Yan Zhu. Dataset distillation by matching training trajectories. In Proceedings of the IEEE/CVF Conference on Computer Vision and Pattern Recognition, pp. 4750–4759, 2022.
> >
> > > [5] Zhou, Yongchao, Ehsan Nezhadarya, and Jimmy Ba. Dataset distillation using neural feature regression. Advances in Neural Information Processing Systems 35, pp. 9813-9827, 2022.

---

> > > ### Author Response · Authors · 2023-11-22
> > >
> > > Dear reviewer w1VW,
> > >
> > > I hope this message finds you well. Would you mind sharing with us the reason why you changed your score?
> > >
> > > Thank you for your time and effort in reviewing our paper again.

---

> > > > ### Comment · Reviewer_w1VW · 2023-11-22
> > > > **Post-rebuttal**
> > > >
> > > > Thanks for the response. After reading the reviews and the responses, my concern about the method's contribution remains. The other reviewers also raise some important issues. Overall, I think this paper has made contributions and helpful discussions, however,  the whole paper's writing and method design, verification of the experiment can be further improved before acceptance.

---

### Official Review · Reviewer_wW36 · 2023-10-29

**Soundness:** 2 fair
**Presentation:** 3 good
**Contribution:** 2 fair
**Rating:** 5
**Confidence:** 5

**Summary:**

This paper presents a set of techniques for training networks on distilled datasets given a fixed (usually small) backbone. The networks for training are not constrained to be the same as those used in the distillation process. The authors argue that the "architecture overfitting" will happen in the distillation process.  To mitigate this, the authors propose a set of techniques to improve the evaluation process and obtain better performance. The authors conduct experiments on two baseline methods and show improved test accuracy.

**Strengths:**

1. The paper is generally written in good quality and easy to follow.
2. The idea is straight-forward and seem to be effective.
3. The topic is timely and important.

**Weaknesses:**

Questions:

1.  I am not sure whether it is really the "overfitting". Why this is an overfitting? Is there any evidence to show that the performance already saturate on one backbone, but decrease on the other backbones? It could be the case that the distillation process is even "underfit" on a given architecture - and the performance drop during "transferring" (i.e., distill with one backbone and then evaluate on the others) could be just some normal "generalization error". I would suggest to use the terminology more carefully to avoid such an confusion, or provide evidence to support the terminology.

2. From what I am understanding, this work actually alters the evaluation process (the process of training a model on the distilled image set). Augmenting the training process is not novel at the high level as many previous works have already done so. A lot of augmentation to architectures (like DropOut and DropPath etc. ) and data (MixUp, CutMix, AutoAugment, etc. ) could be applied here and I think that can serve as potential baselines.

3. Table 2 provides lower results for MTT baselines. From the original paper, MTT reaches 65.3/71.6 accuracy on CIFAR-10, while in Table 2 the authors reported 63.6 and 70.2. I am curious why there is a performance gap since it may create invalid performance advantages for the proposed techniques.


4. Section 4.2 is interesting, but I think it is off-topic. The paper is trying to solve the so-called "architectural overfitting" but it will not happen when there is no actual "fitting" process. Therefore, I think Contribution 3 is do not count and the contribution bullets should be adjusted.

**Questions:**

See the above section. My score will be updated accordingly after the rebuttal.

---

> ### Author Response · Authors · 2023-11-20
>
> Dear reviewer wW36,
>
> Thanks for your constructive suggestions and insightful comments. In response to your questions, we offer the following point-to-point answers:
>
> 1. \>\>\> I am not sure whether it is really the "overfitting". Why this is an overfitting? Is there any evidence to show that the performance already saturates on one backbone, but decreases on the other backbones? It could be the case that the distillation process is even "underfit" on a given architecture - and the performance drop during "transferring" (i.e., distill with one backbone and then evaluate on the others) could be just some normal "generalization error". I would suggest to use the terminology more carefully to avoid such a confusion, or provide evidence to support the terminology.
>
>     **Reply:** We are sorry about the confusion. From the perspective of evaluation, we argue that the architecture overfitting is consistent with the classical definition of overfitting: a model suffers from overfitting if the gap between its performance on the training set and the test set is big. In our cases, the training accuracy on the distilled dataset is always 100\% for all network architectures, but the test accuracy on the real dataset is lower, so overfitting definitely occurs as we see generalization gaps. Moreover, the generalization gap is bigger when we train larger networks on the distilled dataset than when we use a small network with the same architecture as the training network. We call such difference *architecture overfitting*, meaning the additional overfitting arising from the architecture difference between the training network and the test network in dataset distillation. From the dataset perspective, we agree with the reviewer that the distilled dataset is "unfit" to the original large dataset given a network architecture. We believe that the distilled dataset loses some useful information from the original large dataset, leading to noticeable performance degradation when training models on the distilled dataset. Nevertheless, it is important to note that this paper primarily focuses on examining the network architecture perspective. In this context, we utilize the term *architecture overfitting* to describe the phenomenon under investigation.
>
> 2. \>\>\> From what I am understanding, this work actually alters the evaluation process (the process of training a model on the distilled image set). Augmenting the training process is not novel at the high level as many previous works have already done so. A lot of augmentation to architectures (like DropOut and DropPath etc.) and data (MixUp, CutMix, AutoAugment, etc. ) could be applied here and I think that can serve as potential baselines.
>
>     **Reply: We have added more experiments in the revised manuscript in Table 5 of Appendix B.1**, where we adopt DropOut, DropPath, MixUp and CutMix as baselines and evaluate their performance with the same setting in the ablation study (ResNet18, FRePO, CIFAR10, IPC=10). Note that AutoAugment only supports uint8 data, the pixel value can only be in \{0, 1, ..., 255\}. However, distilled data are float32 and their pixel values are neither bounded by [0, 1] nor [0, 255]. As a result, we do not include AutoAugment in the comparison. Similar to AutoAugment, we already adopted 2-fold data augmentation in our method, which also randomly samples 2 augmentations from a pool of augmentations. In addition, from the results shown in Table 2 (w/o DP \& KD v.s. Full), where *w/o DP \& KD* already uses better data augmentation and better optimizer, we can expect that pure data augmentation will not outperform our method. The results of the comparison with different baselines are reported in the table below. We can observe that simple DropPath and DropOut with a constant keep rate can even deteriorate the performance. In addition, MixUp and CutMix only lead to marginal performance improvement, which further demonstrates the effectiveness of our method.

---

> ### Author Response · Authors · 2023-11-20
> **Continue**
>
> 3. \>\>\> Table 2 provides lower results for MTT baselines. From the original paper, MTT reaches 65.3/71.6 accuracy on CIFAR-10, while in Table 2 the authors reported 63.6 and 70.2. I am curious why there is a performance gap since it may create invalid performance advantages for the proposed techniques.
>
>     **Reply:** We carefully checked the hyperparameters of MTT and found the only difference is that the learning rate they adopted in the evaluation is a learnable parameter, which is learned during the distillation process. To ensure a fair comparison with other methods, we disabled this mechanism and used the initial learning rate they defined before distillation, i.e., 0.01. In addition, as pointed out in the original MTT paper [1], the test accuracies fluctuate a lot with different random seeds. Moreover, the accuracies obtained with our hyper-parameters in Table 2 of the manuscript are in the fluctuation range reported in the MTT paper at some runs, e.g., 64.8 / 71.5. Finally, we need to point out that if we compare MTT with the baselines as listed in [1], the fluctuation of the test accuracies does not challenge the performance superiority of MTT.
>
> 4. \>\>\> Section 4.2 is interesting, but I think it is off-topic. The paper is trying to solve the so-called "architectural overfitting" but it will not happen when there is no actual "fitting" process. Therefore, I think Contribution 3 does not count and the contribution bullets should be adjusted.
>
>     **Reply:** As pointed out in the answer to Question 1, architecture overfitting accounts for the additional generalization gap when we train models of different architectures on the distilled dataset, compared with training on small datasets. In section 4.2, we demonstrate that our methods contribute more performance improvement in the case of training on distilled dataset than in the case of training on sub-sampled dataset, so it is useful to mitigate *architecture overfitting*. In addition, section 4.2 validates the effectiveness of our methods in sub-sampled dataset, which indicates the versatility of our method.
>
> ***Table 1: Comparison between our method and baselines. Note that p in DropPath and Dropout denotes the keep rate and alpha in MixUp and CutMix is the parameters $\alpha$ and $\beta$ in beta distribution, where $\alpha$ and $\beta$ are the same.***
> | Method |  Test Accuracy |
> | ------ |  ---- |
> | Baseline in the paper | 55.6|
> |DropPath (p=0.5) | 46.2 |
> |DropOut (p=0.5) | 44.3 |
> |MixUp (alpha=0.5) | 57.6 |
> | CutMix (alpha=0.5) | 56.9  |
> | **Ours** | **66.2** |
>
>
> > [1] George Cazenavette, Tongzhou Wang, Antonio Torralba, Alexei A Efros, and Jun-Yan Zhu. Dataset distillation by matching training trajectories. In Proceedings of the IEEE/CVF Conference on Computer Vision and Pattern Recognition, pp. 4750–4759, 2022.

---

> > ### Author Response · Authors · 2023-11-22
> >
> > Dear reviewer wW36,
> >
> > I hope this message finds you well. I am writing to remind you that we have responded to your comments and would like to know if our reply addresses your concerns. If you have any further questions or concerns, we are glad to have further discussion with you.
> >
> > Thank you for your time and effort in reviewing our paper again.

---

> ### Comment · Reviewer_wW36 · 2023-11-22
> **Response**
>
> For the first question I am partly agree with the authors. The overfitting **does** happen **in the retraining (evaluation) phase**. But I do not agree with how the authors divide the scope.
> When I first read this paper I had the first impression that the authors are trying to address the overfitting **in the distillation phase**.  This leads me to confusion why architecture overfitting even happens. So either make it more clear or change slightly the writing. This is also the main reason hinders me from providing a higher score.
>
> For the MTT part, could the authors provide the results with learnable learning rates? I think that is the *standard* "MTT" people are trying to compare with. Or mark it explicitly and explain why you adopt a fixed learning rate in the paper.
>
> Scores have been adjusted. New experiments are helpful. Thank you.

---

> > ### Author Response · Authors · 2023-11-22
> >
> > Dear reviewer wW36,
> >
> > Thanks for your reply. We would like to address your remaining concerns as follows:
> >
> > 1. Thanks for pointing this out. We will take your comments into consideration and make the necessary changes to improve the quality of our work.
> >
> > 2. MTT does not provide the final "learned" learning rates in the released checkpoints, the only way to obtain these is to reproduce the whole distillation process, which is for us unrealistic at this moment. We also added an explanation for why we adopted a fixed learning rate in Appendix C of the updated paper.
> >
> > Please let us know if you have any further questions or concerns.

---

### Official Review · Reviewer_MbH7 · 2023-10-30

**Soundness:** 4 excellent
**Presentation:** 4 excellent
**Contribution:** 2 fair
**Rating:** 5
**Confidence:** 5

**Summary:**

This paper proposes a series of methods to improve the performance of models trained by synthetic datasets obtained through dataset distillation. Especially, the paper focuses on models with different architectures from those seen during dataset distillation. The proposed strategies include DropPath with a three-phase keep rate, knowledge distillation, and some other designs on learning rate, optimizer, and data augmentation. Experiments demonstrate that the proposed method improves the performance of training on different architectures using synthetic datasets in dataset distillation by a large margin.

**Strengths:**

1. This is the first work that explicitly focuses on the cross-architecture issue in dataset distillation. The topic itself is very meaningful since dataset distillation has been demonstrated to be easily overfitting to a single architecture used in training.
2. The writing is coherent and easy-to-follow.
3. The experiments are sufficient to demonstrate the performance of the proposed strategies and the advantages over existing baselines.

**Weaknesses:**

1. I do not think the cross-architecture generalization problem should be addressed this way. The authors do not modify the process of obtaining synthetic datasets during dataset distillation. Instead, they modify the training strategies **given** synthetic datasets. In dataset distillation, we should definitely focus on the former and should not make any assumptions on how users should use provided synthetic datasets for **downstream** training. What I want to see is actually a thorough algorithm for the process of dataset distillation that can improve the cross-architecture performance following the original evaluation protocols. The current form does not really enhance dataset distillation. The improvement is from the strategies of using distilled datasets.
2. If the paper focuses on how to use distilled datasets, the problem can be cast to some more classical problems, like how to avoid overfitting, synthetic-to-real generalization, few-shot learning, etc. The authors fail to make a broader discussion.
3. The technical novelty is limited because the authors only provide strategies with minor designs that can empirically improve the performance. Without sufficient analysis, the principles of how the proposed methods work are unclear, which results in limited scientific value.
4. The proposed methods are somewhat complicated. For example, they assume users would apply a three-phase keep rate during training with DropPath, which introduces lots of hyper-parameters and makes the pipeline complex and less robust.

**Questions:**

Please refer to Weaknesses for details.

---

> ### Author Response · Authors · 2023-11-20
>
> Dear reviewer MbH7,
>
> Thanks for your insightful and constructive comments. In response to your questions, we offer the following point-to-point answers:
>
> 1.  \>\>\> I do not think the cross-architecture generalization problem should be addressed this way. The authors do not modify the process of obtaining synthetic datasets during dataset distillation. Instead, they modify the training strategies **given** synthetic datasets. In dataset distillation, we should definitely focus on the former and should not make any assumptions on how users should use provided synthetic datasets for **downstream** training. What I want to see is actually a thorough algorithm for the process of dataset distillation that can improve the cross-architecture performance following the original evaluation protocols. The current form does not really enhance dataset distillation. The improvement is from the strategies of using distilled datasets.
>
>      **Reply:** We agree with the reviewer that proposing novel dataset distillation methods may help mitigate architecture overfitting and is definitely a direction for exploration. However, we argue that it is also meaningful to improve the training strategies using the distilled dataset. Our argument is supported by the following points:
>
>     - The overfitting issue studied in this work is not **solely** caused by dataset distillation because a similar overfitting phenomenon is also observed when we train models on tiny real datasets. For example, the results in Section 4.2 indicate overfitting happens when we use a tiny sub-sampled training set to train deep neural networks. Our proposed methods can also improve the performance in this context.
>
>     - Following the first point, from the results in Section 4.1 and 4.2, we observe that training on distilled datasets has better performance than training on sub-sampled datasets. However, the performance gaps between the small and big networks are also larger when training on the distilled datasets. In this context, our proposed methods can induce more performance boosts, so it is more meaningful to apply our methods for distalled datasets, which is also the focus of this paper.
>
>     - In addition, our methods are orthogonal to dataset distillation since our method focuses on **how to use the distilled dataset**, and dataset distillation focuses on **how to obtain the distilled dataset**. These two categories of methods can help improve each other. For example, by adopting our method, we can more comprehensively evaluate the performance on different distilled datasets and thus compare different dataset distillation methods fairer.
>
> 2. \>\>\> If the paper focuses on how to use distilled datasets, the problem can be cast to some more classical problems, like how to avoid overfitting, synthetic-to-real generalization, few-shot learning, etc. The authors fail to make a broader discussion.
>
>     **Reply:** As mentioned above, we argue that discussing our method in the context of dataset distillation is practical. In addition to the distilled dataset, in section 4.2, we also demonstrate the effectiveness of our method when we train deep neural networks on small but real sub-sampled training sets. Although we see bigger performance improvement in applying our method to distilled datasets, its effectiveness on sub-sampled datasets indicates its broader applications to avoid overfitting. However, we argue that the other two cases pointed out by the reviewer are different from our problem settings.
>
>     1. **Synthetic-to-real generalization:** It measures the performance of the models that are trained on the synthetic data but tested on the real data. The goal of this task is to narrow down the gap between the performance on synthetic and real data and thus to make the models more robust to domain shifts [1]. That is to say, the key challenge in generalization lies in addressing the domain shifts rather than insufficient training data or architecture differences as in our work.
>
>     2. **Few-shot learning**: it aims to adapt pre-trained models for new tasks from very few instances. The key to few-shot learning is how to construct a pre-train model on several related tasks with enough training data in the meta-training phase so that it can generalize well to unseen (but related) tasks with just a few examples during the meta-testing phase [2]. However, we focus on training a model from scratch on a small dataset, which is different from few-shot learning settings.
>
>     In our revised manuscript, we add a brief discussion about these related tasks. Customizing our method for them is out of the scope of this paper. We leave them as future works.

---

> > ### Author Response · Authors · 2023-11-20
> > **Continue**
> >
> > 3.  \>\>\> The technical novelty is limited because the authors only provide strategies with minor designs that can empirically improve the performance. Without sufficient analysis, the principles of how the proposed methods work are unclear, which results in limited scientific value.
> >
> >     **Reply:** Although the elements of our method are motivated by some existing techniques such as DropPath and knowledge distillation, we adapt them in the context of dataset distillation. For DropPath, we make it applicable to single-branch networks. We also propose the improved shortcut connection and the three-phase keep rate to boost the performance. For knowledge distillation, use the small network as the teacher network which is contrary to common practices. Regarding the analyses, the ablation studies in the experiments demonstrate the effectiveness and necessity of each of these components. More fundamental analysis of DropPath and knowledge distillation were well analyzed in many works [3 - 5].
> >
> >     We agree that the recipe of our solutions may look straightforward, but **1)** the method is new, and to the best of our knowledge, there is no existing work that use these technique to tackle the architecture overfitting issue of dataset distillation, a key issue of current dataset distillation methods; **2)** the method is well-motivated as demonstrated in section 3, we explained why we need DropPath to decrease the effective depth, three-phase keep rate to stabilize training and knowledge distillation to utilize small network to guide the training of large network. For practitioners, we offer a generally applicable and easy-to-tune (as demonstrated in the ablation study) solution to solve an important issue of dataset distillation and indicate the potential of applying our method to other scenarios as well, such as training on the real but tiny dataset. We believe we do not necessarily need complicated methods to solve a problem, utilizing existing techniques wisely can also generate solutions.
> >
> > 4. \>\>\> The proposed methods are somewhat complicated. For example, they assume users would apply a three-phase keep rate during training with DropPath, which introduces lots of hyper-parameters and makes the pipeline complex and less robust.
> >
> >     **Reply:** Actually, we do not intensively tune the hyper-parameters (HPs). In the ablation study, we evaluate the performance of different HPs. **Upon the result of the reviewer, additional ablation studies on HPs of three-phase keep rate were added**. The result is reported below and in Figure 4 (b) and (c). Compared to the significant performance improvement brought by our method, if we choose HPs in an **appropriate but wide** range, the performance fluctuation caused by different HPs is negligible. For example, when the final keep rate ranges from 0.6 to 1.0, KD weight ranges from 0.3 to 0.9 and KD temperature ranges from 1 to 10, the performance fluctuation is with 1 percentage point. In addition, although lower minimum keep rates and longer periods of decay induce better performance, they also mean a longer training phase. We set them to 0.5 and 500 to balance the efficiency and performance, respectively. Despite that, the performance fluctuation caused by these two HPs is still insignificant. Therefore, we can demonstrate that our training method is robust to different HPs. In addition, our selected HPs perform well in different settings, including different distilled data, IPCs and test networks.
> >
> > ***Table 1: Ablation study on the keep rate in the final phase.***
> > |  Final Keep Rate   | 0.6 | 0.7 | 0.8 | 0.9 | 1.0 |
> > |  ----  | ----  | ----  | ----  | ----  | ----  |
> > | Test Accuracy  | 66.2 | 66.3 | **66.6** | 66.3 | 66.4 |
> >
> >
> > ***Table 2: Ablation study on the period of decay.***
> > |  Period of Decay   | 100 | 200 | 300 | 400 | 500 | 600 | 700 | 800 | 900 | 1000|
> > |  ----  | ----  | ----  | ----  | ----  | ----  | ----  | ----  | ----  | ----  | ----  |
> > | Test Accuracy  | 63.5 | 65.3 | 66.5 | 66.3 | 66.6 | 66.8 | 66.9 | **67.3** | 67.2 | 67.1 |
> >
> > > [1] Chen, Wuyang, et al. "Contrastive syn-to-real generalization." arXiv preprint arXiv:2104.02290 (2021).
> >
> > > [2] Parnami, Archit, and Minwoo Lee. "Learning from few examples: A summary of approaches to few-shot learning." arXiv preprint arXiv:2203.04291 (2022).
> >
> > > [3] Hahn, Sangchul, and Heeyoul Choi. "Understanding dropout as an optimization trick." Neurocomputing 398 (2020): 64-70.
> >
> > > [4] Wang, Hongjun et al. “CamDrop: A New Explanation of Dropout and A Guided Regularization Method for Deep Neural Networks.” Proceedings of the 28th ACM International Conference on Information and Knowledge Management (2019): n. pag.
> >
> > > [5] Phuong, Mary, and Christoph Lampert. "Towards understanding knowledge distillation." International conference on machine learning. PMLR, 2019.

---

> > ### Comment · Reviewer_MbH7 · 2023-11-22
> >
> > I sincerely thank the authors for the responses. However, I still think the methods presented in the paper is somewhat empirical and heuristic. I tend to maintain my original score.

---

> > > ### Author Response · Authors · 2023-11-22
> > >
> > > Dear reviewer MbH7,
> > >
> > > Thank you for your feedback. We appreciate your time and effort in reviewing our paper. We understand your concerns regarding the methods presented in the paper. We will take your comments into consideration and make the necessary changes to improve the quality of our work.

---

### Official Review · Reviewer_Mz7n · 2023-11-01

**Soundness:** 4 excellent
**Presentation:** 4 excellent
**Contribution:** 3 good
**Rating:** 6
**Confidence:** 3

**Summary:**

The submission considers the problem of architecture overfitting in dataset distillation. In dataset distillation, one derives a small synthetic dataset from a given larger, real dataset (along with a “training” network) that captures the learnable properties of the real dataset and can be used to train other “test” networks more efficiently to hopefully achieve similar performance as achievable with the real dataset. An issue that has been noticed in prior work is that when the training and testing networks differ more in architecture, the distilled synthetic dataset starts being less useful, in terms of the task performance.

The submission suggests two main modifications to training the test networks. (1) Since the training networks used for distillation tend to be somewhat shallow, while the downstream test networks are intended to be larger, one heuristic is to train the test networks with a form of regularization that simulates shallower networks. In particular, dropping layers randomly can simulate a form of shallowness (with linear transforms to account for dimension matching). (2) Since it is known that performance drops upon moving to a larger network, one can further attempt to improve training of the test networks by using knowledge distillation to match the predictive distributions of the test network to the smaller training network acting as teacher.

These two tricks seem useful, illustrating significant improvements on existing benchmarks, for a choice of existing dataset distillation methods.

**Strengths:**

Originality: While the techniques discussed aren’t particularly original, the exploration in the context of dataset distillation seems unique, to my knowledge.

Quality: The paper seems to be of reasonable quality overall.

Clarity: The paper is reasonably clear, as long as the reader is already familiar with how dataset distillation works. (I wasn’t very familiar, and needed to skim past literature to follow the procedure and nomenclature.)

Significance: My naive understanding about the practical relevance of dataset distillation is that it has the potential to enable training very large models efficiently, by minimizing the dataset size, as well as applications where training for longer is a bottleneck (as in continual learning and neural architecture search). Modifying the training procedure of the test networks with sensible regularizations that enable this transfer can be quite significant in practice.

**Weaknesses:**

I sense no major weaknesses in the submission. Some minor questions remain which are listed in the following section.

**Questions:**

1. I’m not sure I followed the reasoning behind the scaling of the DropPath output maps. It would be nice to have a derivation in the Appendix for how this scaling matches the expectations from training to test.

2. There’s a statement that architecture overfitting occurs due to depth and not width — has this been recognized in existing work?

3. In the Three-Phase Keep Rate section on page 4, it is said that the variance increases as p increases: isn’t the variance maximal at p = 0.5?

4. There seem to be some hyper-parameters involved, such as the values shaping the shape of the three-phase cycle and the temperature in knowledge-distillation. How are these hyper-parameters tuned? The experiments suggest direct evaluation on test set performances.

---

> ### Author Response · Authors · 2023-11-20
>
> Dear reviewer Mz7n,
>
> Thank you for your constructive comments and suggestions, especially mentioning our work is unique in the context of dataset distillation and is significant in practice.
> For your questions, we offer our point-to-point responses as follows:
>
> 1.  \>\>\> I'm not sure I followed the reasoning behind the scaling of the DropPath output maps. It would be nice to have a derivation in the Appendix for how this scaling matches the expectations from training to test.
>
>      **Reply:** Eq. (1) on page 4 shows that $\mathtt{DropPath}(\mathbf{x}) = \frac{m}{p}\cdot \mathbf{x},\quad m=\mathtt{Bernoulli}(p)$, where $p \in [0, 1]$ denotes the keep rate, $m = \mathtt{Bernoulli}(p) \in \{0,1\}$ outputs 1 with probability $p$ and 0 with probability $1-p$. We consider the module output $\mathbf{y} = \mathtt{DropPath}(\mathbf{x})$ in the training phase, then the expectation of $\mathbf{y}$ given $\\mathbf{x}$ is $\mathbb{E}(\mathbf{y}) = p \cdot \frac{1}{p} \cdot \mathbf{x} + (1 - p) \cdot \frac{0}{p} \cdot \mathbf{x} = \mathbf{x}$. In the test phase, DropPath is disabled, so the module output is simply $\mathbf{x}$ and consistent with the expectation in the training phase. If there is no scaling factor $1 / p$ in Eq. (1) and $p < 1$, the expectation of the module outputs in the training and test phases will be different, which leads to performance degradation.
>
> 2.  \>\>\> There is a statement that architecture overfitting occurs due to depth and not width. Has this been recognized in existing work?
>
>     **Reply:** First, to clarify, we do not claim that the width has no impact on architecture overfitting. We use the architectures evaluated in existing works, including [1, 2], and observe that the architecture overfitting occurs and becomes more severe when the models get deeper (from 3-layer CNN to 50-layer ResNet). We do not discuss the impact of width in this context.
>
> 3.  \>\>\> In the Three-Phase Keep Rate section on page 4, it is said that the variance increases as p increases: isn't the variance maximal at p = 0.5?
>
>     **Reply:** We are sorry about the confusion, and we agree with your insightful comment. For a Bernoulli distribution, the variance is indeed the largest when $p=0.5$. We have updated the corresponding statement (please check the text highlighted in blue in our revised manuscript). The key point of three-phase keep rate is to control the **effective depth** of the model by the value of $p$. In the early phase of training, the model is underfitting, stochastic architecture brings optimization challenges for training, so we turn off DropPath by setting the keep rate $p = 1$ in the first few epochs to ensure that the network learns meaningful representations. We then gradually decrease $p$ to decrease the effective depth of the model and thus to narrow down the performance gap between the deep sub-network and the shallow training network until the keep rate reaches the predefined minimum value after several epochs. In the final phase of training, we decrease the architecture stochasticity by increasing the value of $p$ to a higher value to ensure training convergence.
>
> 4.  \>\>\> There seem to be some hyper-parameters involved, such as the values shaping the shape of the three-phase cycle and the temperature in knowledge-distillation. How are these hyper-parameters tuned? The experiments suggest direct evaluation on test set performances.
>
>     **Reply:** In Section 4.3, we conduct ablation study to evaluate the performance under different hyper-parameters (HPs) when using FRePo-generated distilled dataset, IPC = 10 and ResNet18 as the test network. **More ablation studies on HPs of three-phase keep rate have been added in the revised manuscript**. The results are reported below and in Figure 4 (b) and (c). We can conclude that the performance fluctuation by different HPs in an **appropriate but wide** range is negligible, compared with the significant performance improvement brought by our method. For example, when the final keep rate ranges from 0.6 to 1.0, KD weight ranges from 0.3 to 0.9 and KD temperature ranges from 1 to 10, the performance fluctuation is with 1 percentage point. In addition, although lower minimum keep rates and longer periods of decay induce better performance, they also mean a longer training phase. We set them to $0.5$ and $500$ to balance the efficiency and performance, respectively. Despite that, the performance fluctuation caused by these two HPs is still insignificant. In addition, our results in Section 4.1 demonstrate that the selected HPs by ablation studies also perform quite well under other different settings, including different distilled datasets, IPs and test networks. In summary, many experimental observations indicate that our training method is robust to different HPs.

---

> > ### Author Response · Authors · 2023-11-20
> > **Supplementary**
> >
> > ***Table 1: Ablation study on the keep rate in the final phase.***
> > |  Final Keep Rate   | 0.6 | 0.7 | 0.8 | 0.9 | 1.0 |
> > |  ----  | ----  | ----  | ----  | ----  | ----  |
> > | Test Accuracy  | 66.2 | 66.3 | **66.6** | 66.3 | 66.4 |
> >
> >
> > ***Table 2: Ablation study on the period of decay.***
> > |  Period of Decay   | 100 | 200 | 300 | 400 | 500 | 600 | 700 | 800 | 900 | 1000|
> > |  ----  | ----  | ----  | ----  | ----  | ----  | ----  | ----  | ----  | ----  | ----  |
> > | Test Accuracy  | 63.5 | 65.3 | 66.5 | 66.3 | 66.6 | 66.8 | 66.9 | **67.3** | 67.2 | 67.1 |
> >
> > > [1] George Cazenavette, Tongzhou Wang, Antonio Torralba, Alexei A Efros, and Jun-Yan Zhu. Dataset distillation by matching training trajectories. In Proceedings of the IEEE/CVF Conference on Computer Vision and Pattern Recognition, pp. 4750–4759, 2022.
> >
> > > [2] Zhou, Yongchao, Ehsan Nezhadarya, and Jimmy Ba. Dataset distillation using neural feature regression. Advances in Neural Information Processing Systems 35, pp. 9813-9827, 2022.

---

> > ### Comment · Reviewer_Mz7n · 2023-11-22
> > **Thanks for the response**
> >
> > I think what was confusing me about the DropPath is that at training time, the two choices are x or a convblocks(x) while at test time it’s the convblocks(x) part, if I understood right. It’s unclear to me if the scaling by p can actually match the expectations without also taking the expected value of the convblocks output into account (which seems quite non-trivial). As a heuristic, it is fine.
> >
> > Regarding the hyper-parameters, while the initial settings seem intuitively set, and display some robustness, it is better practice to tune hyper-parameters on a held-out set. Otherwise, one can suspect that the initial settings are always likely to require direct access to test set to find stable regions.
> >
> > Overall, I think the submission has some potential, and retain my ratings. I also agree with some of the criticisms raised by the other reviews.

---

> > > ### Author Response · Authors · 2023-11-22
> > >
> > > Dear reviewer Mz7n,
> > >
> > > Thanks for your reply. We would like to address your remaining concerns as follows:
> > >
> > > 1. We would like to clarify that for single-branch networks, DropPath works as you described. However, for multi-branch networks, at training time, the two choices should be **x** or **x + convblocks(x)**  while at test time it is **x + convblocks(x)**. As for the scaling, we follow PyTorch's implementation of the DropOut function https://pytorch.org/docs/stable/generated/torch.nn.Dropout.html, where the scaling factor is also adopted, but $p$ in their implementation denotes discard rate, so the scaling factor is $1/(1-p)$. We also tend to agree that this is a heuristic approach.
> > >
> > > 2. Thanks for pointing this out. We will take your comments into consideration and make the necessary changes to improve the quality of our work.
> > >
> > > Please let us know if you have any further questions or concerns.

---

### Author Response · Authors · 2023-11-20
**General Response**

We appreciate all reviewers' constructive and insightful comments and their hard work. First, we would like to list the revisions to the manuscript as follows, which are also highlighted in blue in the updated paper.

1. In Introduction (page 2), we add a brief discussion of synthetic-to-real generalization and few-shot learning, we compare them with the settings in this paper.
2. In Section 3.1 (page 4 and 5), we add more details about the motivation and analysis of DropPath and three-phase keep rate.
3. In Section 4.3 (page 8 and 9), we add ablation studies on different final keep rates and different periods of decay, and plot the results in Figure 4 (b) and (c). In addition, the curves of different (knowledge distillation) KD weights and different KD temperatures are moved to Appendix B.6 (page 17).
4. We add a discussion about the effect of the scaling factor $1/p$ in DropPath in Appendix A.2 (page 14).
5. We add more baselines for comparison, including DropOut, DropPath, MixUp and CutMix, in Appendix B.1 (page 15).
6. We revise the statement in the implementation details of DropPath (Appendix C, page 19) to elaborate that the stabilization epoch and total epoch are calculated based on the period of decay.

In addition, we make a general response to the question mentioned by more than one reviewer:

-  \>\>\> There are some hyperparameters in our method. How are these hyperparameters tuned? Do they make the pipeline complex and less robust?

   **Reply:** Actually, we do not intensively tune the hyper-parameters (HPs). In the ablation study, we evaluate the performance under different HPs. **Moreover, additional ablation studies on HPs of three-phase keep rate were added**. The result is reported below and in Figure 4 (b) and (c) of our revised manuscript. Compared to the significant performance improvement brought by our method, if we choose different HPs in an **appropriate but wide** range, the performance fluctuation caused by different HPs is negligible. For example, when the final keep rate ranges from 0.6 to 1.0, KD weight ranges from 0.3 to 0.9 and KD temperature ranges from 1 to 10, the performance fluctuation is smaller than 1 percentage point. In addition, although lower minimum keep rates and longer periods of decay induce better performance, they also mean a longer training phase. We set them to 0.5 and 500 to balance the efficiency and performance, respectively. Despite that, the performance fluctuation caused by these two HPs is also marginal. Finally, the selected HPs also perform well in different settings, including different distilled data, different IPCs and different test networks. Therefore, we can demonstrate that our training method is robust to different HPs.

***Table 1: Ablation study on the keep rate in the final phase.***
|  Final Keep Rate   | 0.6 | 0.7 | 0.8 | 0.9 | 1.0 |
|  ----  | ----  | ----  | ----  | ----  | ----  |
| Test Accuracy  | 66.2 | 66.3 | **66.6** | 66.3 | 66.4 |


***Table 2: Ablation study on the period of decay.***
|  Period of Decay   | 100 | 200 | 300 | 400 | 500 | 600 | 700 | 800 | 900 | 1000|
|  ----  | ----  | ----  | ----  | ----  | ----  | ----  | ----  | ----  | ----  | ----  |
| Test Accuracy  | 63.5 | 65.3 | 66.5 | 66.3 | 66.6 | 66.8 | 66.9 | **67.3** | 67.2 | 67.1 |

---

### Meta-Review · Area_Chair_f1zi · 2023-12-10

**Metareview:**

The submission tackles the observed issue with architecture overfitting in dataset distillation. Dataset distillation is the process of creating a small synthetic dataset from a larger real dataset, which can then be used to train a new neural network, in place of the larger real dataset. A practically observed issue is that the gap in transfer accuracy is high when the networks used to distill the dataset (net1) and later train (net2) are different.

The authors propose a few methods to overcome this gap - 1) Add regularization to the training of net2 to make it more similar to net1, 2) Use distillation losses in training of net2, using net1 as teacher, 3) Better choice of optimizer and training procedure.
The reviewers in consensus felt that the methods presented were heuristic and lacked sufficient novelty and insight. It is not clear if this is a major contribution to the process of dataset distillation, given that most the contributions are for the subsequent network training. As one reviewer mentioned during the AC-reviewer discussion, "This work primarily emphasizes post-training network improvement rather than the distillation process itself, which places it in the category of potentially general network training techniques for performance enhancement."

The AC would also like to point out that the writing can be improved - the abstract, summary of contributions in the introduction, and the conclusion, all do not make it clear what the "exact contributions" really are. They only mention in vague terms that "we propose approaches to mitigate the issue". The authors are recommended to make these sections more informative and to the point.

**Justification For Why Not Higher Score:**

This submission received 4 ratings, which were negative leaning with 3 x 5 and 1 x 6. The reviewer that gave it a 6 mentioned in the AC-reviewer discussion that they do not feel strongly enough to champion the submission. The reviewers believe that the contributions of the work is under the bar for acceptance, and I do not have sufficient reason to overturn the consensus.

**Justification For Why Not Lower Score:**

N/A

---

### Decision · Program_Chairs · 2024-01-16

Reject